



# Seismic Deformation of Himalayan Glaciers Using Synthetic Aperture Radar Interferometry

Sandeep Kumar Mondal[1,2], Rishikesh Bharti[1,2], and Kristy F Tiampo[3]

[1]Technology Innovation Hub (TIH), Technology Innovation and Development Foundation (TIDF), Indian Institute of
Technology Guwahati, Assam, India
[2]Earth System Science and Engineering Division, Department of Civil Engineering, Indian Institute of Technology Guwahati,
Assam, India
[3]Department of Geological Sciences, University of Colorado Boulder, Boulder, CO, USA

*Correspondence to*: Rishikesh Bharti (rbharti@iitg.ac.in)

**Abstract.** The Himalayan belt, formed due to Cenozoic convergence between the Eurasia and Indian craton, acts as a
storehouse of large amounts of strain that results in large earthquakes that occur from the western to the eastern Himalayas.
Glaciers also occur over a major portion of the high-altitude Himalayan region. In this study, we attempt to understand the
impact of earthquakes on and around Himalayan glaciers in terms of vertical deformation and coherence change. Eight
earthquake events of various magnitudes and hypocenter depths occurring in the vicinity of Himalayan glacial bodies have
been analyzed using C-band Sentinel1-A/B synthetic aperture radar (SAR). Differential interferometric SAR (DInSAR)
method is applied to the pre-and co-seismic single look complex (SLC) SAR imagery to capture glacial surface deformation
potentially related to earthquake occurrence. The influence radius of each earthquake (Mw>4.5) is defined using shake maps.
For the lower magnitude earthquakes (Mw<4.5), influence radii are computed using the linear relationship between influence
radii and magnitudes of past earthquakes. The mean glacial displacement varies from -38.9 mm to -5.4 mm for the 2020 Tibet
earthquake (Mw 5.7) and the 2021 Nepal earthquake (Mw 4.1). However, small glacial and ground patches processed
separately for vertical displacements reveal that the glacial mass shows much greater seismic displacement than the ground
surface. This indicates potential site-specific seismicity amplification properties of glacial bodies that need additional studies.
Reduction in co-seismic coherence around the glaciers is observed in some cases, indicative of possible changes in the glacial
moraine deposits and/or vegetation cover. The effect of two different seismic events (the 2020 and 2021 Nepal earthquakes)
with different hypocenter depths but the same magnitude at almost equal distances from the glaciers is assessed; a shallow
earthquake is observed to result in a larger impact on glacial bodies in terms of vertical displacement. Earthquakes may induce
glacial hazards such as glacial surging, ice-avalanches and failure of moraine-/ice-dammed glacial lakes. This research can
assist in identifying areas at risk and provide valuable insights for planning and implementing measures for disaster risk
reduction in the near future.



## 1 Introduction

Glaciers are the perennial storehouse of freshwater in high altitudes/low latitudes. Apart from the polar cryosphere, the Himalayan cryosphere has the largest glacial coverage accounting for an area of ~33,000 km$^2$ (Dyurgerov and Meier, 1997). Glaciers behave differently in different seasons and glacier dynamics (flow velocities) vary seasonally due to enhanced melting rates in the ablation period and reduced melting in the accumulation period (Zwally et al., 2002; Joughin et al., 2008; Scherler

and Strecker, 2012; Satyabala, 2016). Glacial dynamics also include surface mobility, glacial facies, changes in the areal extent and elevation profile that spaceborne/airborne remote sensing techniques have been studied extensively (Berthier et al., 2005; Kumar et al., 2011; Bhambari et al., 2012; Bhardwaj et al., 2015; Thakur et al., 2016, 2017). Glaciological investigations conducted from field-based and satellite-driven studies have shown that the Himalayan glaciers behave in synchronization with various meteorological parameters such as temperature and precipitation resulting in retreat and negative mass balance

(Wagnon et al., 2007; Dobhal et al., 2008; Bhambari et al., 2011; Azam et al., 2014; Pratap et al., 2016). These glaciers are located in the challenging, inaccessible, and seismically-prone areas of the eastern and western Himalayas.

Earthquake impact assessment a crucial task that must be conducted after seismic events. The most common approach for determining earthquake magnitude and aftereffects is the analysis of ground-based seismometer datasets. However, due to their limited deployment in high-altitude glaciated regions, it is difficult to study certain earthquake events that occur close to

those glacial bodies. Therefore, a number of geophysicists and seismologists have supplemented such ground-based methods with geodetic and satellite remote sensing-based approaches (Li et al., 2021).

Routine assessment of damage and continuous monitoring through the synoptic acquisition of data and analysis is essential to studying natural hazards, particularly earthquakes in the remote glaciated regions of Himalayan ranges which are sudden, unpredictable and are capable in activating other hazards such as avalanches, glacial lake outburst floods and landslides.

Therefore, extensive monitoring at high temporal and spatial resolution is essential using techniques such as differential global positioning system (DGPS), photogrammetry and optical remote sensing (Guilfold et al., 2008). Radar technology, due to its cloud penetrating capacity, is advantageous for all-weather glaciological monitoring and proves to be more suited for studying the rugged and valley-adorned terrain of mountainous regions such as the Himalayas (Goldstein et al., 1993; Strozzi et al., 2002). The approach to measuring surface changes with the help of SAR can be divided into amplitude-based and phase-based

methods using either L- or C-band radar. Radar interferometry has served as an effective technique to study the Himalayan cryosphere and associated natural hazards such as avalanches (Leinss et al., 2020; Mondal and Bharti, 2022a), slope failures (Mondal and Bharti, 2022b), glacial lake outburst floods (Cochachin et al., 2017; Scapozza et al., 2019), landslides (Riedel et al., 2008; Zhang et al., 2018; Mondal et al., 2022) and earthquakes (Thomas, 2021; Li et al., 2021; Mondal and Bharti, 2022c). The pros and cons of these microwave bands derives from the sensor wavelength. The shorter the wavelength, the better the

spatial resolution, whereas the longer the wavelength, the better the surface penetration (Howell et al., 2018; Jelenek and Kopačková-Strnadová, 2021). L-band SAR has been used to study soil moisture, geomorphological features and larger amplitude tectonic motions (Chaparro et al., 2018; Zhang et al., 2019; Fernandez-Carrillo et al., 2019), X-band SAR systems





have been used to track landslides and lava flows at finer scales (Kubanek et al., 2017), and C-band SAR systems provide a compromise between both wavelengths (Atzori et al., 2013; Wang et al., 2018; Howell et al., 2018).

There are various techniques in microwave remote sensing to measure glacial surface velocity, such as SAR interferometry (Kumar et al., 2011), Multiple Aperture Interferometry (MAI) (Gourmelen et al., 2011), and offset tracking (Strozzi et al., 2002). Where intensity and coherence tracking measures two-dimensional displacement in azimuth and slant range direction, the DInSAR only estimates displacement in slant range direction (Strozzi et al., 2002). For tracking glacial surface displacement in time, DInSAR is an important technique, provided there exists a short temporal and geometrical baseline

(Mondal and Bharti, 2022c). This technique employs a set of differential interferograms acquired for the same location over a period of time to derive a time series of temporal variations in deformation (Berardino et al., 2002). DInSAR is a very effective method for analyzing surface deformations due to subsidence, landslides, earthquake-triggered slope failures, glacial velocity, and other natural hazards (Peltzer and Rosen, 1995; Massonnet et al., 1995; Mouginot et al., 2012; Samsonov et al., 2020; Devaraj and Yarrakula, 2020; Crosetto et al., 2020). For the Himalayan glaciers, offset tracking and DInSAR are the most

common and widely used techniques (Kumar et al., 2009, 2013; Varugu et al., 2015; Thakur et al., 2018; Lal et al., 2018). In earthquake research, DInSAR has been used to estimate the co-seismic deformation field (Yang et al., 2019; Wang et al., 2018). In recent times, DInSAR has become a preferred technique for deriving precise surface deformation of glaciers on a temporal basis (Yasuda and Furuya, 2013; Milillo et al., 2017). Here, an attempt is made to capture the impact of seismic events on and around glacial bodies situated in different parts of the Himalayan region using the DInSAR technique.

**2 Study area**

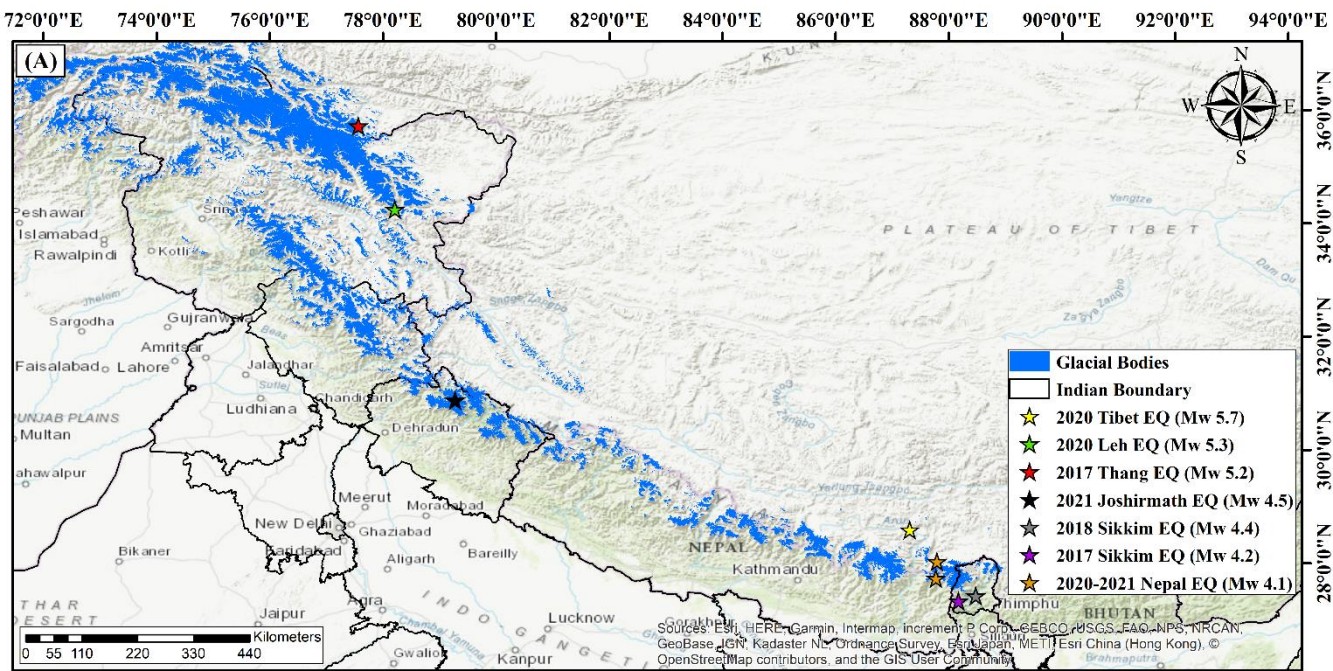



**Figure 1: Study area map showing the Himalayan glaciers and selected earthquake epicenters (Sources: Esri, HERE, Garmin International, and others as mentioned on the map).**

The collision between the Indian and Eurasian plates during the Cenozoic period resulted in the formation of two major orogenic features, the Tibetan plateau and the Himalaya-Hindu Kush mountains (Rajendran et al., 2017). The Himalayan orogeny falls under the category of active orogenic belt with geological and geomorphic structures of the Quaternary-Holocene age (Thakur, 2013). Three principal thrusts, namely the main central thrust (MCT) (Hubbard and Harrison, 1989), main boundary thrust (MBT) (Meigs et al., 1995), and Himalayan frontal thrust (HFT) (Thakur et al., 2007), developed progressively in the early Miocene, late Miocene and Quaternary period respectively. The Himalayan arc and its contiguous emergent structures are seismically active and have resulted in very large seismic events in the recent past, including devastating earthquakes such as the 1897 Shillong (Mw 8.1), the 1905 Kangra (Mw 7.8), the 1934 Bihar-Nepal (Mw 8.2), and the 1950 Assam (Mw 8.6) earthquakes. In the present study, eight earthquake events of different magnitudes, hypocenter depth, and epicenter locations that have occurred in the recent past and are close to the Himalayan glacial bodies considered here, as shown in Fig. 1 and Table 1.

**Table 1: Details of earthquake events in this study**

| Events | Date/ UTC Time | Epicenter Location | Seismic Magnitude (Mw) | Hypocenter Depth (km) |
|---|---|---|---|---|
| Tibet Earthquake | 20th March 2020/ 01: 33: 15 | 28.590°N 87.308°E | 5.7 | 10 |
| Leh Earthquake | 25th Sept 2020/ 10: 57: 13 | 34.249°N 78.206°E | 5.3 | 10 |
| Thang Earthquake | 6th Dec 2017/ 23: 29: 03 | 35.724°N 77.562°E | 5.2 | 74.3 |
| Joshimath Earthquake | 23rd May 2021/ 19:01:39 | 30.876°N 79.271°E | 4.5 | 10 |
| Sikkim Earthquake | 17th June 2018/ 15: 07: 37 | 27.420°N 88.474°E | 4.4 | 49.8 |
| Sikkim Earthquake | 16th May 2017/ 22: 13: 19 | 27.327°N 88.171°E | 4.2 | 10 |
| Nepal Earthquake | 8th Nov 2020/ 21: 44: 14 | 27.730°N 87.771°E | 4.1 | 10 |
| Nepal Earthquake | 11th Jan 2021/ 13: 59: 49 | 28.028°N 87.789°E | 4.1 | 35 |

## 3 Data and methods

### 3.1 Data

Sentinel-1A/B SAR datasets were used in this study. Developed by the European Space Agency (ESA), Sentinel-1A/B is a dual satellite configuration that acquires data in the microwave C-band ($\lambda$: 5.6cm, f: 5.405 GHz) (Winsvold et al., 2018). Both the satellites lie on the same orbit plane with a combined and individual temporal resolution of 6 days and 12 days, respectively. Several data items are provided by Sentinel-1A/B, including Level-1 Ground Range Detection (GRD), Level-2 Ocean (OCN), and Level-0, Level-1 Single Look Complex (SLC). It also has various modes of operation, namely Wave (WV), Strip Map (SM), Extra Wide Swath (EW), and Interferometric Wide Swath (IW) modes. The IW captures data in a 250 km swath having



a spatial resolution of 5m×20m. IW mode acquires data in three sub-swaths with the help of Terrain Observation with
Progressive Scans SAR (TOPSAR). Each sub-swath data is an aggregate of a series of bursts, and each burst is processed as a
separate SLC file (ESA, 2016). In this study, we used pre- and coseismic pairs of SLC datasets, as shown in Table 2. Sentinel-
2 Level 1C (L1C) multispectral instrument (MSI) data from the Nepal earthquake region of 17th Jan 2021 also was used to
observe the vegetation coverage through false color composite (FCC) image. The L1C MSI product comprises orthorectified
tiles covering an area of 100 km$^2$ (ESA, 2015). Radiometric measurements for the L1C products are available in Top of
Atmosphere (TOA) reflectance, where the datasets have been resampled with constant Ground Sampling Distance (GSD) of
60, 20 and 10 m depending upon the native resolution of various spectral bands (ESA, 2015). Here, green (543-578 nm) and
red (658-680 nm) bands at 10 m and near-infrared (785-899 nm) band in 20 m resolution have been used to generate a False
Color Composite (FCC) image of the zone of study. The boundary of the Himalayan glacial bodies has been adopted from the
Randolph glacial inventory, RGI 6.0 (RGI Consortium, 2017).

**Table 2: Sentinel-1A/B C-band SLC images used in this study**

| Sentinel 1 SLC Datasets | | | | | | |
|---|---|---|---|---|---|---|
| *Year-Earthquake* | *Pre-Earthquake Datasets* | | *Co-Earthquake Datasets* | | *Beam Mode-Sub-swath/Path* | *Frame/Flight Direction* | *Polarization* |
| | *Primary* | *Secondary* | *Primary* | *Secondary* | | | |
| 2020-Tibet Earthquake | 04-03-2020 | 16-03-2020 | 16-03-2020 | 28-03-2020 | IW-1/121 | 496/Descending | VV+VH |
| 2020-Leh Earthquake | 07-09-2020 | 19-09-2020 | 19-09-2020 | 01-10-2020 | IW-1/136 | 476/Descending | VV+VH |
| 2017-Thang Earthquake | 15-11-2017 | 27-11-2017 | 27-11-2017 | 09-12-2017 | IW-1,2/129 | 112/Ascending | VV+VH |
| 2021-Joshirmath Earthquake | 06-05-2021 | 18-05-2021 | 18-05-2021 | 30-05-2021 | IW-1,2/63 | 487/Descending | VV+VH |
| 2018-Sikkim Earthquake | 30-05-2018 | 11-06-2018 | 11-06-2018 | 23-06-2018 | IW-3/12 | 84/Ascending | VV+VH |
| 2017-Sikkim Earthquake | 02-05-2017 | 14-05-2017 | 14-05-2017 | 26-05-2017 | IW-2/48 | 499/Descending | VV+VH |
| 2020-Nepal Earthquake | 25-10-2020 | 06-11-2020 | 06-11-2020 | 18-11-2020 | IW-3/48 | 499/Descending | VV+VH |
| 2021-Nepal Earthquake | 24-12-2020 | 05-01-2021 | 05-01-2021 | 17-01-2021 | IW-3/48 | 499/Descending | VV+VH |

## 3.2 Seismic events and interferometric pair selection

The Himalayan region has experienced a large number of earthquakes of different magnitudes and hypocenter depths
distributed from west to east. However, the primary goal of this study is to understand the impact of earthquake events in terms
of glacial deformation. Therefore, earthquakes based on the magnitude range from Mw 5.7 to Mw 4.1 with varying hypocenter
depths have been selected close to Himalayan glaciers at various locations.

DInSAR for seismic deformation studies relies on selecting SLC data pairs to obtain a good quality co-seismic deformation
field. Since the study is based on understanding the seismic deformation of Himalayan glaciers, interferometric pairs have
been selected to capture complete seismic signal based on (a) the SLC products should provide the best possible areal coverage



within the influence area of an earthquake; (b) the shortest possible temporal baseline; (c) the earliest post-seismic acquisition;
(d) the maximum possible areal coverage around the epicenter location.

These criteria helped minimize decoherence, as the shortest temporal baseline ensures high coherence, especially while investigating low latitude regions, and minimizing the effect of ongoing glacial deformation after the earthquake. The influence area for each earthquake event has been determined using USGS shake maps (https://earthquake.usgs.gov/data/shakemap/). For the earthquake events devoid of shake maps, influence area has been
determined from the shake maps of other earthquakes with similar magnitude and hypocenter depth.

### 3.3 Vertical Deformation

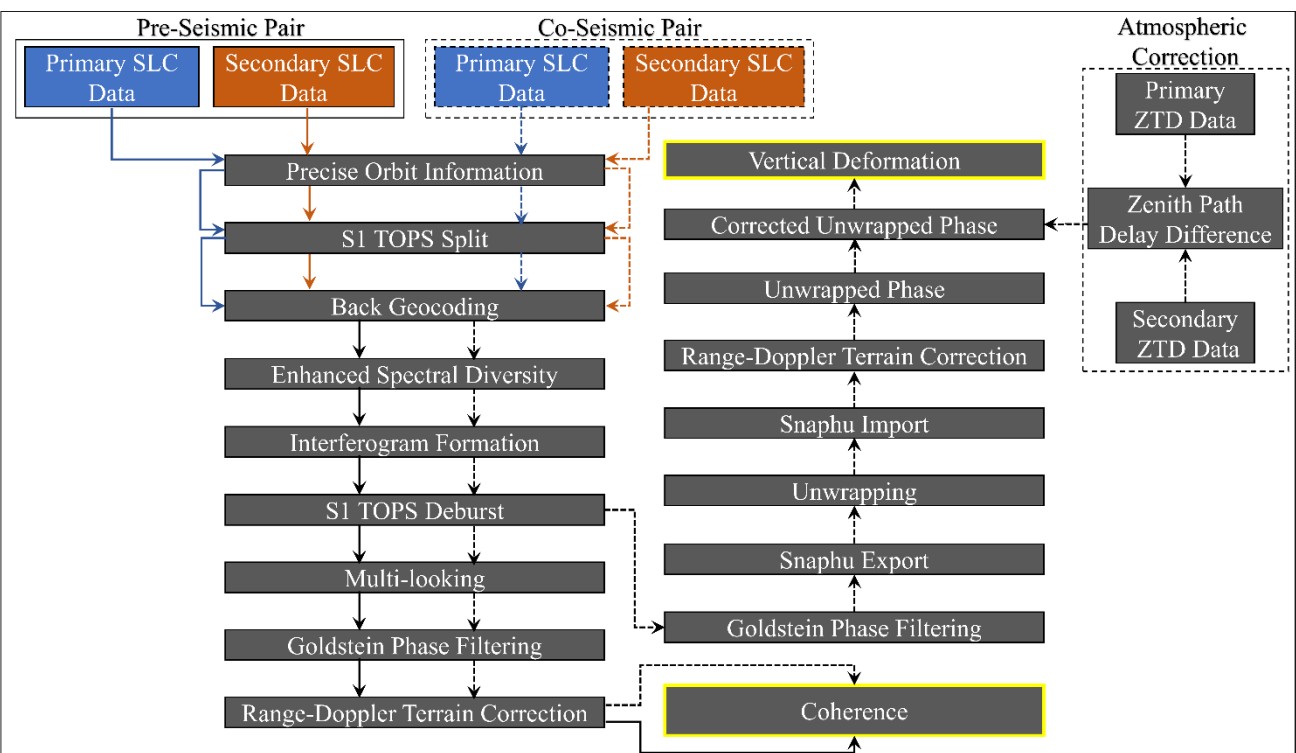

**Figure 2: Process diagram for the estimation of atmospherically corrected vertical deformation and coherence.**

SAR systems such as Sentinel-1A/B are active radar systems operating in the microwave spectrum, transmitting and receiving
pulses from the Earth's surface. These pulses or signals consist of phase and amplitude information. Therefore, in a SAR image, information from each pixel can be treated as a complex number where the real part represents the wave, and the imaginary part corresponds to phase and amplitude values (Moreira et al., 2013). The processing begins with identifying the primary and secondary images for DInSAR analysis, as shown in Fig. 2. This involves obtaining the pre-seismic event SLC data as the primary file and the co-seismic data as the secondary file, respectively. The orbit state vectors were improved using precise
orbit information for the SLC products. This was followed by co-registration of primary and secondary images for the same



sub-swath using 30m Shuttle Radar Topography Mission (SRTM) digital elevation model (DEM). For co-registration, the constant range offset was obtained by averaging the range offsets for every burst in the SLC split of IW mode (Varade et al., 2019). Enhanced Spectral Diversity (ESD) was used to determine the constant azimuth offset. These offsets are required for applying azimuth and range corrections for each burst in the SLC split (Yague-Martinez et al., 2016).

Interferograms were generated through standard methods based on phase differences of data pairs. A deburst algorithm was used to integrate the interferogram for every individual bursts into a single sub-swath product (Yague-Martinez et al., 2016). This is followed by topographic phase removal for phase flattening (Zhou et al., 2009). Goldstein phase filtering was applied to increase the signal-to-noise (SNR) ratio, thus enhancing the unwrapping accuracy (Werner and Goldstein, 1998). The flattened interferogram delivers an ambiguous estimation of the relative terrain altitude within $2\pi$ cycles of the interferometric

phase. In the field of radar interferometry, the principal observation is the relative phase signal in line-of-site (LOS), depicted as the $2\pi$-modulus of the absolute phase signal (Thomas, 2021).

Phase unwrapping is the process of adding the appropriate integer multiple of $2\pi$ to flattened interferometric fringes to produce the absolute interferometric phase information from its wrapped product lying in the range of $-\pi$ to $+\pi$ (ESA, 2016). To generate a continuous deformation field, the consecutive fringes in the flattened interferogram need to be unwrapped, which

involves interpolating phase jumps over 0 to $2\pi$. Here, the phase unwrapping of the generated filtered interferogram has been performed in the statistical-cost, network-flow algorithm for phase unwrapping (SNAPHU) (Chen and Zebker 2002). Orthorectification (geocoding) of the unwrapped phase was performed using Range Doppler terrain correction (Hellwich and Ebner 2000). Finally, the vertical displacement from the unwrapped phase has been derived using the following equation:

$$z_{vertical} = \frac{\phi_{unwrapped}\lambda_{SAR}}{-4\pi\cos\theta_{in}} \qquad (1)$$

where $\phi\_{unwrapped}$ represents unwrapped phase interferogram, $\lambda_{SAR}$ is the C-band wavelength of Sentinel-1 SAR instrument (57 mm) and $\theta_{in}$ is the incident angle (Walter, 2014). Division by the cosine of the incidence angle converts the line-of-sight (LOS) displacement to vertical displacement. Sentinel-1 acquiring datasets at a wavelength of 5.56 cm are affected due to atmospheric delay, which can be eradicated using zenith total delay (ZTD) maps as obtained from generic atmospheric correction online service for InSAR (GACOS) (http://www.gacos.net/). GACOS makes use of the iterative tropospheric

decomposition (ITD) model to segregate turbulent and stratified signals from tropospheric total delays (Yu et al., 2018; Li et al., 2018). This helps generate ZTD maps at high spatial resolution used to correct unwrapped interferograms as obtained through phase unwrapping. The difference in ZTD maps obtained for the primary and secondary datasets are deducted from the unwrapped phase to obtain atmospherically corrected unwrapped phase further used to derive vertical displacement.

**3.4 SAR interferometric coherence**

To introduce a comparison between two SAR scenes and determine the coherence, the SAR pair needs to be precisely aligned, resampling the secondary scene to the primary scene. It is generally preferred that the baseline of imagery is low (<200 m) along with zero doppler SAR pair geometry (Olen and Bookhagen, 2018). Interferometry can be seen as the complex coherence



between corresponding samples of SAR pair estimated from the absolute value of correlation coefficient (Tamkuan and Nagai, 2017; Natsuaki et al., 2018). The complex coherence can be explained using Eq. (2).

$$\gamma_{complex} = \frac{X\{a,b^*\}}{\sqrt{X\{|a^2|\}.X\{|b^2|\}}} \tag{2}$$

where X denotes the expectation operator, a and b are zero mean circular Gaussian variables, and * represents the complex conjugate. The coherence value can be obtained from the complex coherence magnitude, as shown in Eq. (3).

$$\gamma = |\gamma_{complex}| = \frac{\frac{1}{n}\Sigma_1^n P.S^*}{\sqrt{\frac{1}{n}\Sigma_1^n P.S^*}.\sqrt{\frac{1}{n}\Sigma_1^n P.S^*}} \tag{3}$$

where n denotes the kernel size of the sliding window. *P* and *S* are primary and secondary SAR scenes, respectively. In a
coherence product, the value ranges between 0 to 1 where 1 stands for perfect coherence which is rarely observed in the nature. Coherence is sensitive to changing pixel information (phase or amplitude). Changes in the surface's backscatter properties, surface elevation or dielectric characteristics can result in the loss of coherence or decorrelation (Zebker et al., 1992). Thus, changing coherence can be helpful in the temporal and spatial estimate of when and where changes have occurred on the ground surface. However, the type or rate of change cannot be assessed (Olen and Bookhagen, 2018). The overall decorrelation
captured from a pair of SAR scenes can be seen as a combination of different types of decorrelation (Strozzi et al., 2000) as shown in Eq. (4).

$$\gamma = \gamma_t \gamma_m \gamma_s \gamma_d \gamma_a \gamma_k \tag{4}$$

where $\gamma_t$ represents thermal decorrelation as a result of uncorrelated noise within the radar sensor (Liu et al., 2014), $\gamma_m$ is the decorrelation due to mis-registration of two SAR scenes (Franceschetti and Lanari, 2018), $\gamma_s$ represents spatial decorrelation
due to large baselines (Hoffman, 2007). Due to large baselines, there exists a significant difference in the incident angle of SAR pair resulting in coherence reduction (Zebker and Villasenor, 1992). In addition, $\gamma_s$ depends on the topography of the area under investigation. The slope facing the sensor shows significant decorrelation with an increase in slope angle due to the layover effect and foreshortening. On flat topography, the effect of $\gamma_s$ can be corrected using common band filtering provided the range resolution is compromised (Hoffman, 2007). Similar to $\gamma_s$, $\gamma_d$ (doppler centroid decorrelation) occurs due to large
variation in the squint angle between SAR-pair acquisitions. To avoid the effect of $\gamma_d$, antenna steering or range adaptive azimuth common band filtering can be adopted (Franceschetti and Lanari, 2018). $\gamma_a$ occurs due to different water content in the atmosphere during both SAR acquisitions resulting in artifacts in the interferogram (Gens, 1996; Gupta, 2003). $\gamma_k$ represents temporal decorrelation due to Earth's surface changes resulting from frost and dew cycles, snowfall, and melting around glaciated areas. In the areas covered by vegetation or forest, coherence loss occurs due to wind and plant growth (Wegmüller
and Werner, 1995; Chen et al., 2002).



## 4 Results and discussion

### 4.1 2020 Tibet Earthquake, Mw 5.7

For the purpose of detecting surface deformation from an earthquake event of sufficiently high magnitude (Mw>5), DInSAR is an effective approach (Li et al., 2021). There exists a strong correlation between the magnitude of an earthquake event and

its detectability through SAR interferometry. Earthquake events with larger magnitude create significant surficial deformation that can be captured in the interferogram, and the deformation signal generated due to high-magnitude earthquake differs from other background processes (Li et al., 2021). Another critical factor is the hypocenter depth. The hypocenter depth governs whether an earthquake event of a given magnitude generates surface deformations with sufficient amplitude to be detected through DInSAR. As a general rule, the deeper the earthquake, the smaller the amplitude of the deformation at Earth's surface

(Mellors et al., 2004). This makes shallow seismic events more detectable through co-seismic interferograms.



**Figure 3: Map for 2020 Tibet earthquake (Mw 5.7) showing (A) orthorectified phase interferogram (B) vertical displacement (C) vertical displacement for coherence ≥0.6, coseismic image (D) orthorectified phase interferogram at epicenter location (E) vertical displacement at epicenter location and (F) vertical displacement for coherence ≥0.6, coseismic image.**

In the case of the 2020 Tibet earthquake, with a hypocenter depth of 10 km, lobate deformation with a 10 km wide series of concentric fringes can be seen in the phase image as shown in Fig. 3A and 3D. The center of the concentric fringes represents the earthquake epicenter which lies approximately at a distance of 13.7 km from the USGS epicenter. The vertical displacement around the epicenter lies between 0 and -168 mm. Negative displacement values indicate the subsidence at the surface, with maximum motion away from the epicenter at the DInSAR-derived epicenter location, as shown in Fig. 3B, C, E, and F. In the

case of DInSAR-based seismic deformation studies, the reliability and quality of results generated from unwrapped phase information depend on the co-seismic coherence (Thomas, 2021). The regions of high co-seismic coherence can be considered as the regions of more reliable surface displacement. To enhance the quality of the derived results, vertical displacement of the regions having high co-seismic coherence (≥0.6) has been masked out, as shown in Fig. 3C and F. The influence area of the earthquake as defined from the isoseismal contours of the event shake map

(https://earthquake.usgs.gov/earthquakes/eventpage/us70008cld/shakemap/intensity) and is considered as a circular area up to a distance of 69 km from the USGS epicenter. The mean vertical displacement in the area is found to be -47.3 mm. We observe that there is only subsidence due to the 2020 Tibet earthquake (Mw 5.7). Li et al., 2022 detected the coseismic deformation for the Tibet earthquake using ISCE (InSAR Scientific Computing Environment) (Rosen et al., 2012) with maximum deformation in the order of -126 and -157 mm through ascending and descending tracks respectively. A seismogenic fault

(dipping northeast) was detected with the main rupture zone of $\sim 5 \times 5$ km$^2$ and maximum coseismic quasi-vertical displacement of -164 mm. Fig. 4B represents the vertical displacement of the glacial bodies ranging between 0 and -79 mm with a mean displacement of -38.9 mm. To check for variation of surficial features such as glacial moraine deposits around the glacial bodies, the changes in coherence due to earthquake event was calculated, as shown in Fig. 4A. A reduction in the coherence can be observed from pre-seismic to co-seismic SLC pair in some of the regions around the earthquake epicenter within the

lacustrine deposits of natural lakes (Lake Dingmu) and human settlements (Mondal and Bharti, 2022c). Reduced coherence can be seen around some glacial bodies. However, such changes are not significant as the glacial bodies are situated close to the periphery of the influence area which, is quite far away from the epicenter of the earthquake.



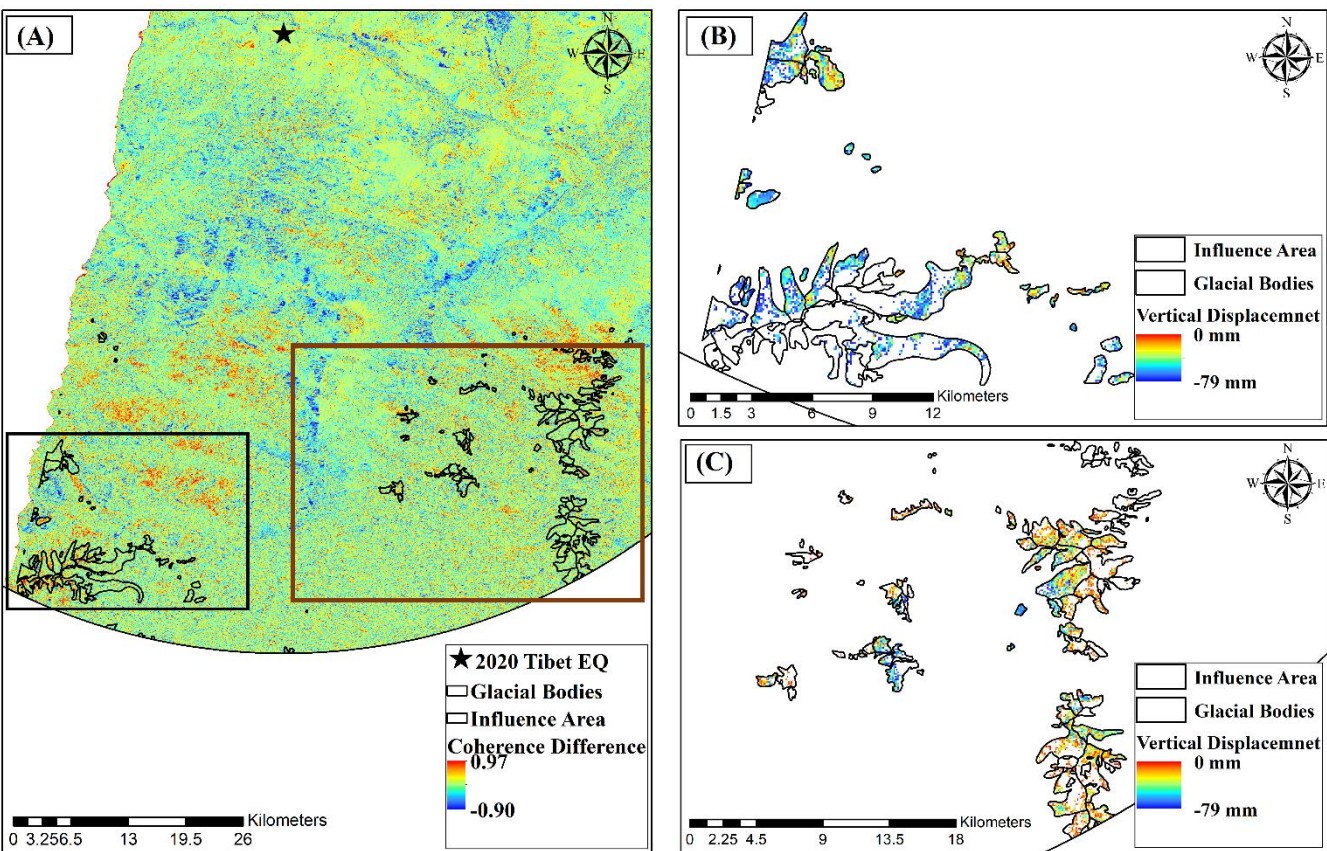

**Figure 4: Map showing orthorectified layer of (A) difference between pre-and co-seismic coherence, (B) and (C) vertical**
**displacement (mm) within glacial bodies derived from unwrapped phase interferogram for 2020 Tibet earthquake (Mw 5.7).**

### 4.2 2020 Leh Earthquake, Mw 5.3 and 2017 Thang Earthquake, Mw 5.2

Figure 5. represents the vertical displacement captured due to the 2020 Leh earthquake (Mw 5.3). Within the influence radius
of 61 km, vertical displacement lies between 31 mm and -33 mm. Unlike the 2020 Tibet earthquake, where there is only
subsidence but no uplift of the surface after the earthquake, regions close to the epicenter location show positive vertical
displacement for the 2020 Leh earthquake, with negative vertical displacements moving away from the epicenter. For the areas
with coherence ≥0.6, the mean vertical displacement is found to be -2.4 mm. Fig. 6A shows the coherence change obtained by
subtracting pre-seismic coherence values from the co-seismic coherence values. Reduction in coherence can be observed
around the glacial bodies. Even though surficial deformations can act as a primary source of decorrelation, other factors such
as decorrelation due to sensor noise from the radar system, geometric decorrelation due to perpendicular baseline component,
and temporal decorrelation due to changes in surficial features and/or vegetation are important (Zebker et al., 1992; Mondal
and Bharti, 2022c). Thus, temporal decorrelation before and during an earthquake event can be attributed to surficial changes
and/or changes in vegetation cover. As Leh is characterized as a high-altitude dry-cold mountainous desert (Krishan, 1996),
such reduced coherence due to an earthquake event around glacial bodies can indicate deformation/alteration of glacial





moraine. However, because of the shape and size of the glacial bodies, it is difficult to unwrap the Fig. 6B represents the
displacement of glacial bodies where the glaciers lying close to the epicenter show positive displacement or uplift. The glacial
bodies far from the epicenter capture subsidence. The mean vertical displacement of glaciers is found to be -2.5 mm. However,
with an influence radius of 61 km, the region of investigation for seismic deformation is large and consists of ground surface
and glacial bodies which are quite different from each other in terms of their material properties. Unwrapping the DInSAR
phase information for the entire influence area of the earthquake may introduce unwrapping errors. Therefore, we investigated
the vertical displacement due to seismic events glaciers and the surrounding ground surface separately, so as to avoid any
unwrapping errors while considering a larger area. Considering small patches of glaciated regions and ground surface
separately (Fig. 7) at almost equal radial distance of 41 km shows that the glacial region experiences negative vertical
displacements of up to -78 mm, which is much greater than the ground displacement, which lies in the range of 18 mm and -
32 mm. The vertical ground displacement shown in the subregion of Fig. 7C lies within the range of seismic displacement as
shown in Fig. 5B. Moreover, where the ground surface shows both uplift and subsidence, the glacial patch shows only
subsidence. One possible reason behind the variation in the seismic deformation of glacial bodies and ground surface could be
the differences in their material properties and dynamic response. Glaciers are huge masses of metamorphosed ice that behave
as a Kelvin Voight (non-Newtonian) material with the ability to flow, which is different from the solid ground surface (Reeh
et al., 2003). As a result, the impact of earthquakes on glacial bodies will be different from that of the ground surface.

For the 2017 Thang earthquake (Mw 5.2), the vertical displacement ranges between 51 mm and -39 mm within the influence
radius of 56 km, with the mean displacement of 2.8 mm for regions masked with coherence ≥0.6 as shown in Fig. 8B. For the
glacial bodies within the influence area, the mean vertical displacement of 3.4 mm is observed (Fig. 9B). From the coherence
difference map (Fig. 9A), reduced co-seismic coherence can be seen within and around the glacial bodies lying within the
influence area. Alike Leh, Thang is also a dry-cold mountain desert devoid of vegetation around the glaciated regions. Reduced
coherence due to a seismic event within and around glacial bodies suggests surficial changes in glacial moraine deposits from
that earthquake. Fig. 10 shows the vertical displacement derived for glacial and ground patches separately. The glacial patches
show very high vertical displacement due to the 2017 Thang earthquake that reaches up to -173 mm and -576 mm as shown
in Fig. 10B and 10C respectively. Whereas, the seismic displacement of the solid earth patches does not exceed -39 mm as
also is observed while considering the entire influence area for seismic displacement as shown in Fig. 8. It also is interesting
to observe that the glacial patch farther from the epicenter and surrounded by larger glacial bodies shows larger vertical
displacement (Fig. 10C) than the glacial patch close to the epicenter (Fig. 10B). One possibility could be site-specific
amplification of the seismic waves which is dependent on the material properties of glacial bodies, including shear modulus,
damping coefficient, and thickness. However, better understanding of these effects would require extensive field- as well as
lab-based studies of the behavior of glacial ice towards seismic loading.




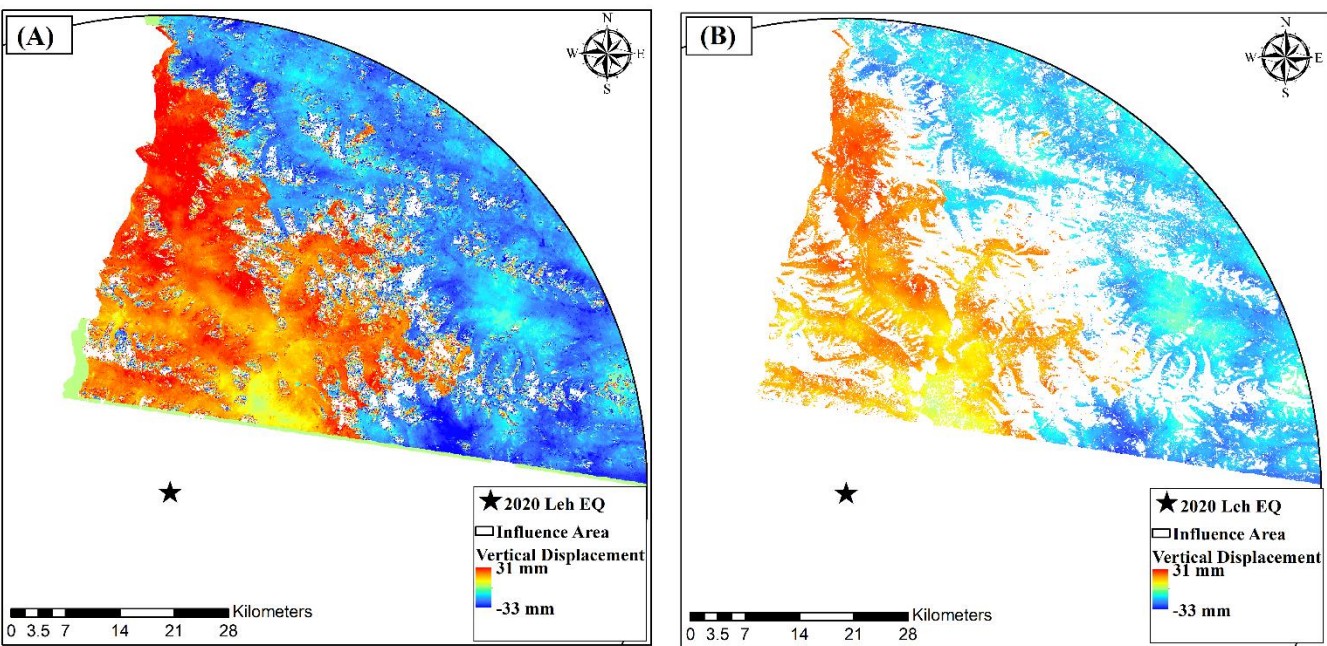


**Figure 5: Map showing (A) orthorectified vertical displacement and (B) vertical displacement with co-seismic coherence ≥0.6 for 2020 Leh earthquake (Mw 5.3).**

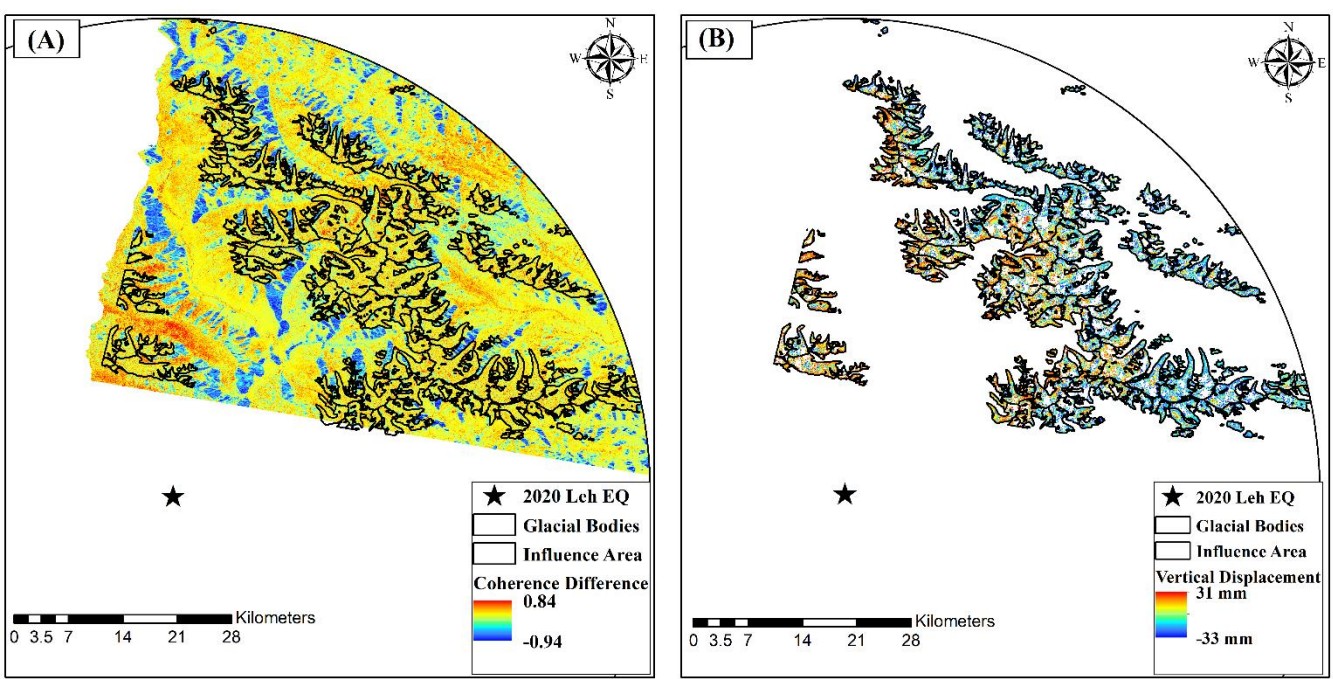

**Figure 6: Map showing (A) difference between pre-and co-seismic coherence and (B) vertical displacement (mm) within glacial**
**bodies derived from unwrapped phase interferogram for 2020 Leh earthquake (Mw 5.3).**



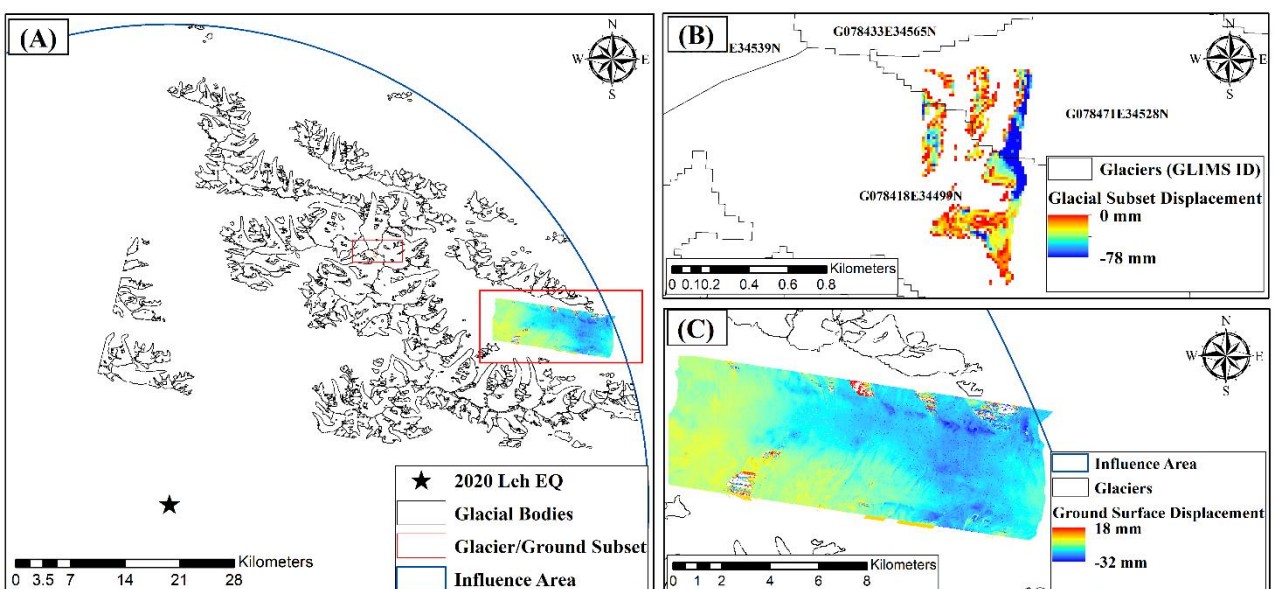

**Figure 7: Map showing (A) glacial and ground subset regions, (B) vertical displacement (mm) within glacial subset and (C) vertical displacement (mm) within ground subset derived from unwrapped phase interferogram for 2020 Leh earthquake (Mw 5.3).**

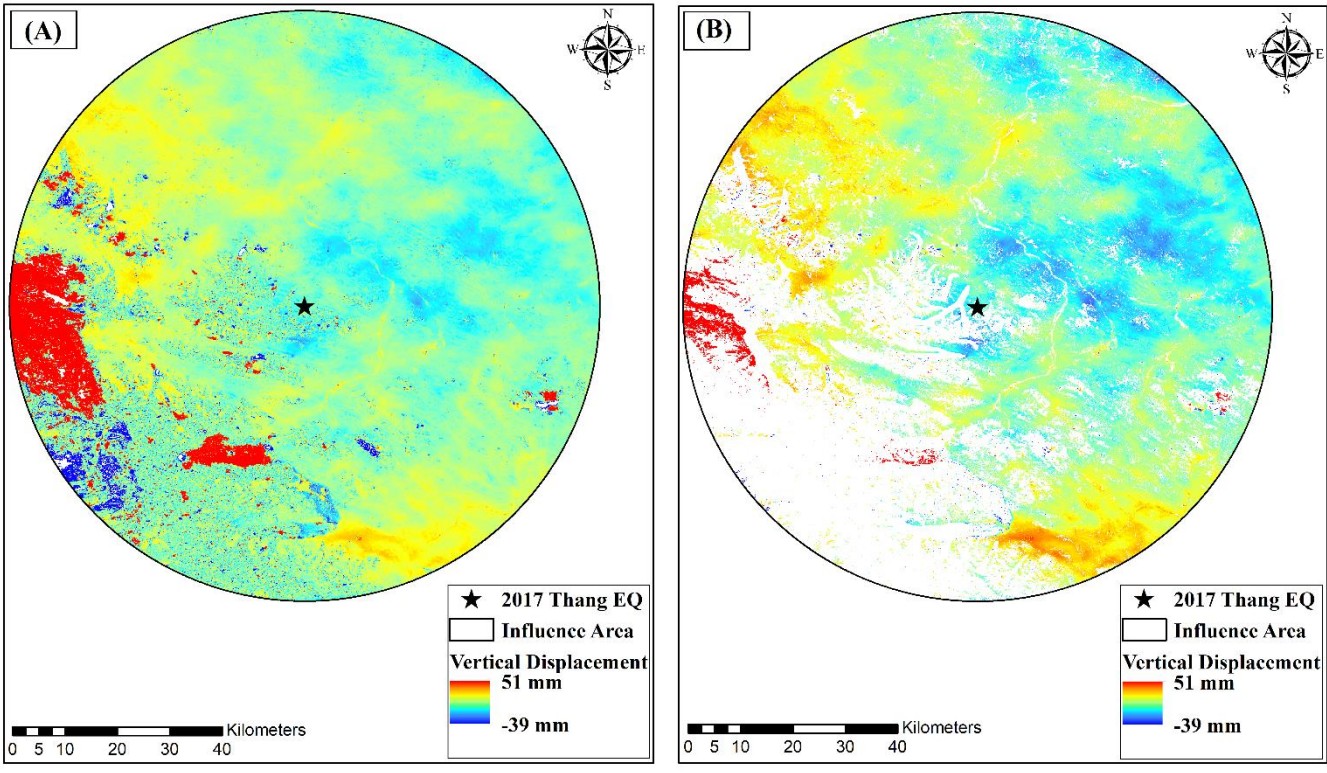

**Figure 8: Map showing (A) orthorectified vertical displacement and (B) vertical displacement with co-seismic coherence ≥0.6 for 2017 Thang earthquake (Mw 5.2).**



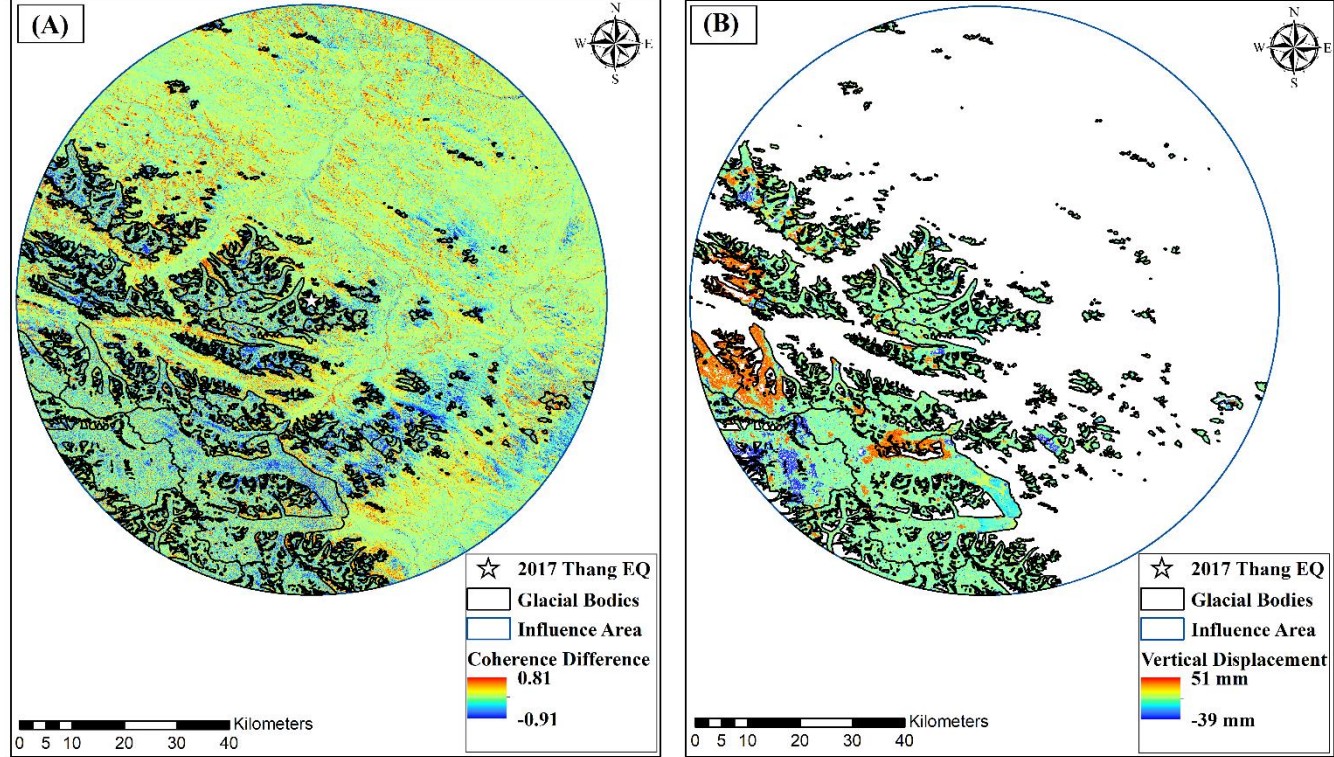

**Figure 9: Map showing (A) difference between pre-and co-seismic coherence and (B) vertical displacement (mm) within glacial bodies derived from unwrapped phase interferogram for 2017 Thang earthquake (Mw 5.2).**





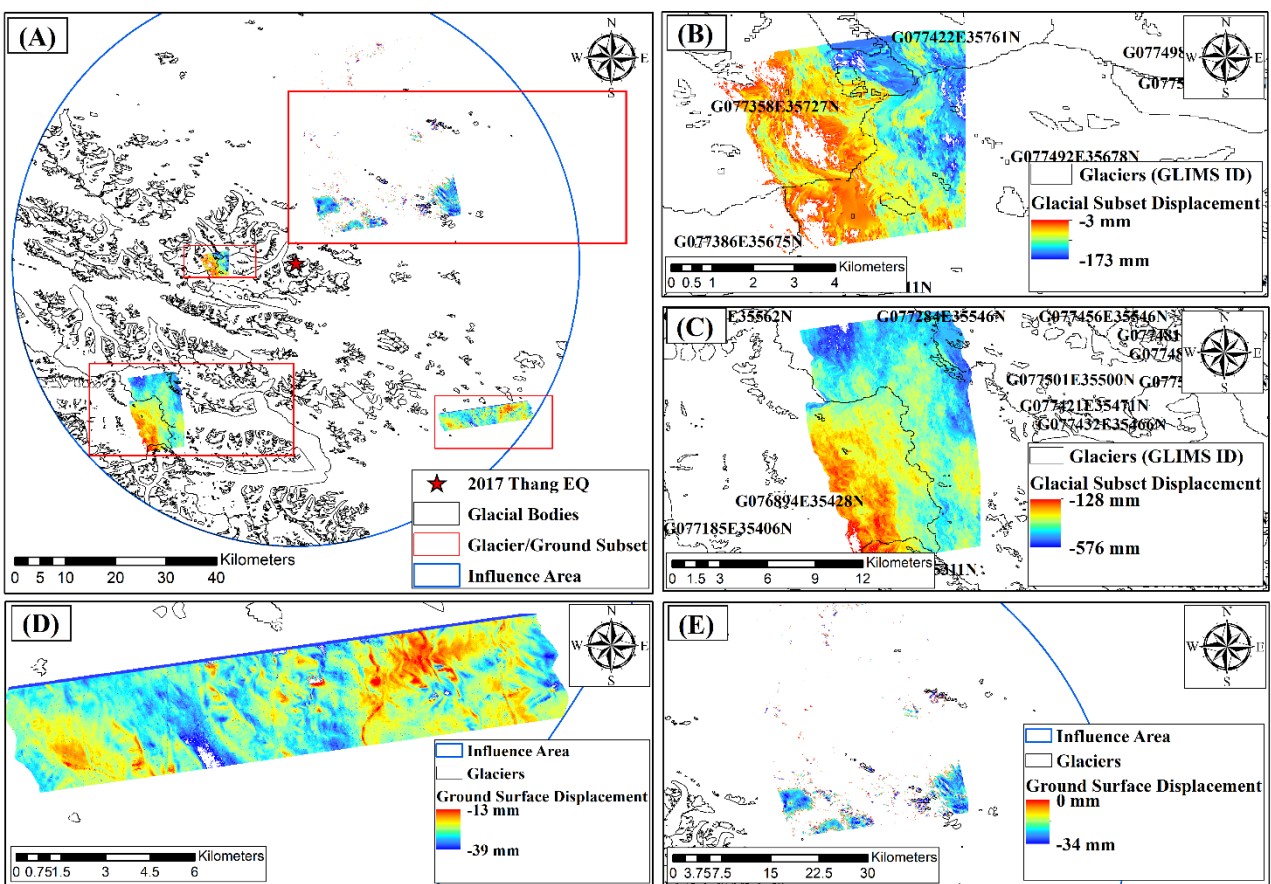

**Figure 10: Map showing (A) difference between pre-and co-seismic coherence and (B) vertical displacement (mm) within glacial bodies derived from unwrapped phase interferogram for 2017 Thang earthquake (Mw 5.2).**

## 4.3 2021 Joshimath Earthquakes, Mw 4.5

The Joshimath earthquake (Mw 4.5) occurred on 23rd May 2021 at a hypocenter depth of 10 km near the Gangotri glacier. Fig. 11 shows the vertical deformation of the Earth's surface within the influence area (54 km). The earthquake resulted in both uplift and subsidence, in the range of 45 mm and -52 mm respectively. However, the mean displacement of -1.7 mm is observed. Earthquake-induced vertical displacement for the regions with co-seismic coherence ≥0.6 shows the mean vertical shift in the order of 0.64 mm. For the glacial bodies, mean vertical displacement of -2.2 mm is observed (Fig. 12C and E). A major region within the influence area shows negative vertical displacement except for some small patches and the northeastern region of the influence area. The majority of glacial bodies are in the situated in the east-west region and very few in the northern side of the influence area. Therefore, the mean vertical displacement of the glacial bodies shows slightly greater subsidence as compared to the mean vertical displacement of the ground surface within the influence area. From the coherence difference map of the influence area, a reduction in the co-seismic coherence can be observed around the glaciated regions along the northern-eastern side of the influence area where earthquake-induced uplift is observed. Fig. 13 shows the seismic





displacement of glacial and solid earth patches. As also was observed for the 2020 Leh earthquake and 2017 Thang earthquake, the glacial patch shows seismic displacement of up to -94 mm due to the 2021 Joshimath earthquake, which is much higher than the vertical displacement of ground patch, of as much as -51 mm.

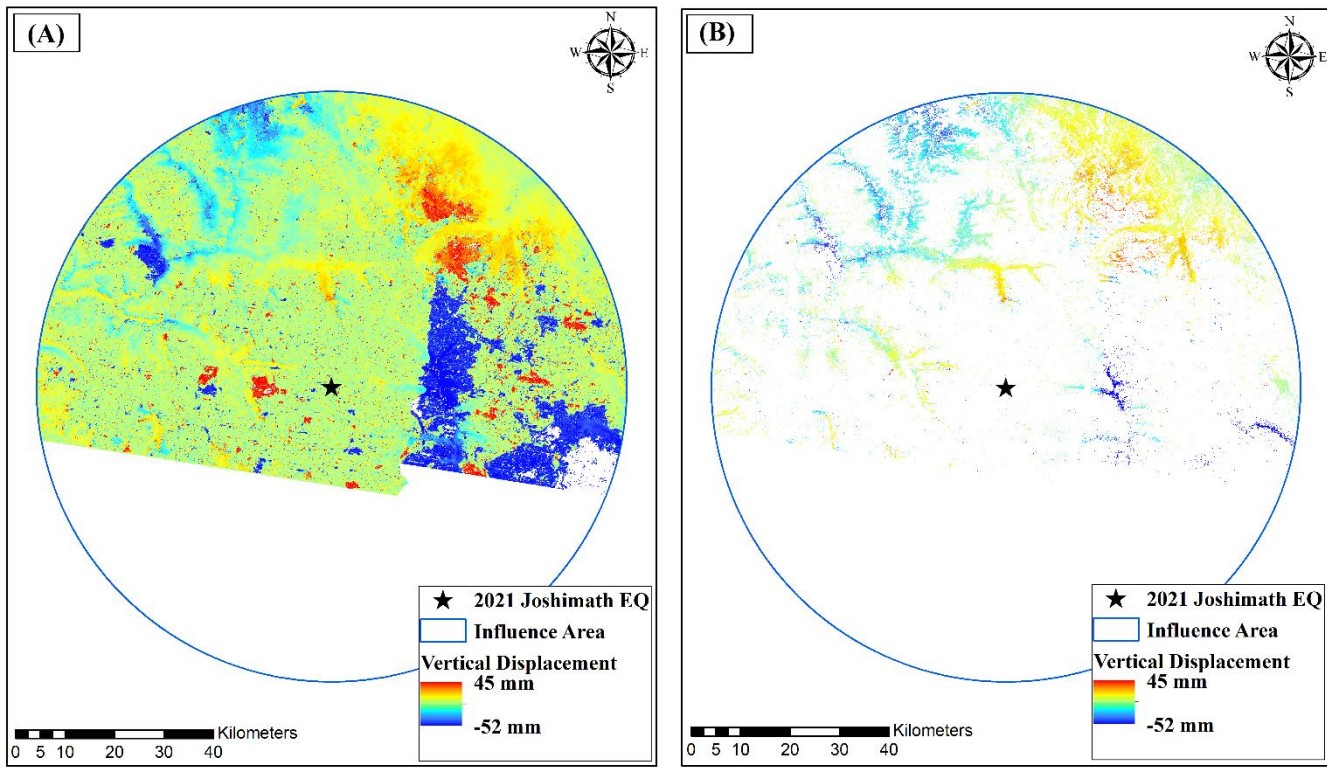

**Figure 11: Map showing (A) orthorectified vertical displacement and (B) vertical displacement with co-seismic coherence ≥0.6 for**
**2021 Joshimath earthquake (Mw 4.5).**



**Figure 12: Map showing (A), (B), (D) difference between pre-and co-seismic coherence and (C), (E) vertical displacement (mm) within glacial bodies derived from unwrapped phase interferogram for 2021 Joshimath earthquake (Mw 4.5).**





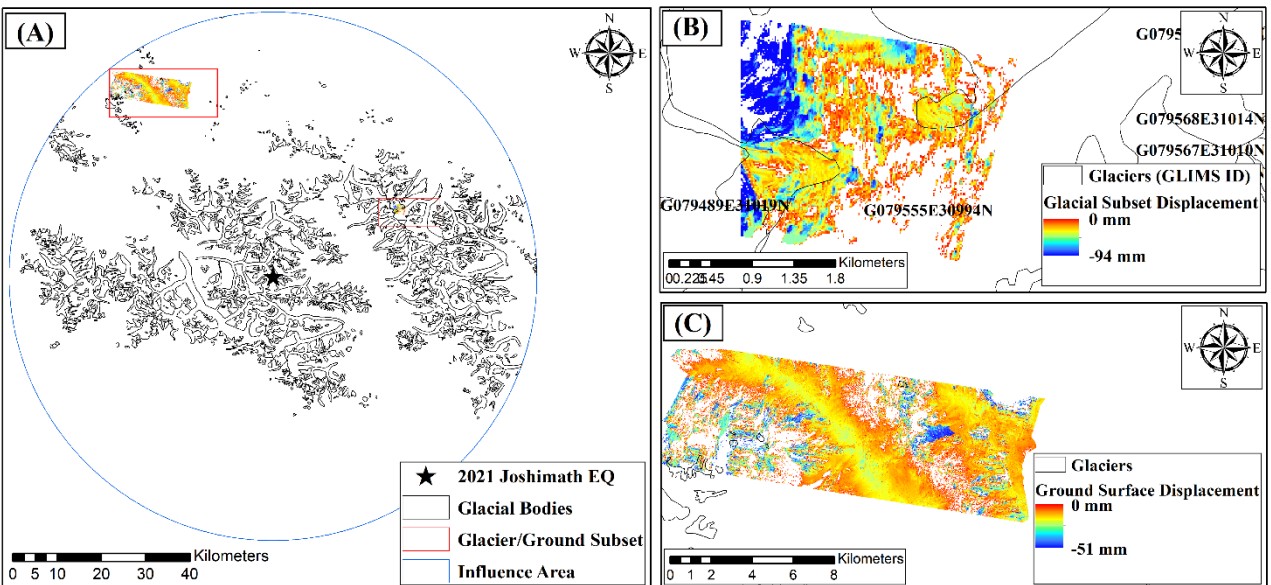

**Figure 13: Map showing (A) glacial and ground subset regions, (B) vertical displacement (mm) within glacial subset and (C) vertical displacement (mm) within ground subset derived from unwrapped phase interferogram for 2021 Joshimath earthquake (Mw 4.5).**

## 4.4 2018 Sikkim Earthquake, Mw 4.4 and 2017 Sikkim Earthquake, Mw 4.2

Sikkim is a northeastern state of India that encompasses the eastern Himalayas and is situated in the high seismic hazard territory (Zone-IV) (IS1893, Bureau of Indian Standards, 2002). Sikkim has experienced moderate seismic impacts in the past.

However, shake maps are not available for the earthquake events of Mw<4.5 within the northeastern Himalayan region. Thus, determining the influence radius of earthquakes is more difficult for lower-magnitude earthquakes due to a lack of isoseismal contours.

As we know, the higher the magnitude (Mw), the greater the surface deformation that can be detected in an interferogram (Li et al., 2021). Also, the higher the hypocenter depth (DH), the smaller the deformation amplitude at the surface (Mellors et al.,

2004). This is well represented through isoseismal contours which govern the influence radius (IR) of an earthquake event in terms of surface deformations. Therefore, the relationship of Mw and DH with IR can be represented as Eq. (5) and Eq. (6) respectively.

$$M_w \propto IR \tag{5}$$

$$D_H \propto \frac{1}{IR} \tag{6}$$

Detection of earthquake impacts on the Earth's surface depends on two major parameters: magnitude and hypocenter depth. The 2017 Sikkim earthquake is triggered at a hypocenter depth of 10 km. In the previous sections, we investigated three other earthquake events of similar hypocenter depth but varying magnitudes, as shown in Table 3. The influence radius of these earthquakes differs linearly as shown in Fig. 14A. Thus, the influence radius of the 2017 Sikkim earthquake is calculated as 46 km. In case of the 2018 Sikkim earthquake, the scenario is slightly different. The earthquake is triggered at a hypocenter




depth of 49.8 km. From past records, earthquakes at exactly the same hypocenter depth have not occurred in the Indian subcontinent. Thus, earthquakes of different magnitudes and hypocenter depths close to 49.8 km have been considered, and their influence radii have been determined from USGS shake maps, as shown in Table 3. However, their relation (*IR vs $M_w$*) is not linear. Therefore, the normalized influence radius (*$IR_N$*) for these earthquake events at a hypocenter depth of 49.8 km has been computed using Eq. (7).

$$IR_N = \frac{D_H}{D_{H'}} \times IR \tag{7}$$

Here, $D_{H'}$ represents the hypocenter depth of 2018 Sikkim earthquake for which $IR_N$ of each earthquake is calculated. This develops a linear relationship between *IR* and $M_w$ as shown in Fig. 14B. Thus, the normalized influence radius of the 2018 Sikkim earthquake is computed as 17 km.

**Table 3: Earthquakes and their respective magnitudes and influence radii**

| Earthquakes with hypocenter depth of 10 km | | |
|---|---|---|
| Event/ Date | $M_w$ | IR (km) |
| Tibet Earthquake/ 20th March 2020 | 5.7 | 69 |
| Leh Earthquake/ 25th Sept 2020 | 5.3 | 61 |
| Joshirmath Earthquake/ 23rd May 2021 | 4.5 | 54 |
| Sikkim Earthquake/ 16th May 2017 | 4.2 | 46 |
| Nepal Earthquake/ 8th Nov 2020 | 4.1 | 39 |
| Earthquakes with varying hypocenter depth close to 49.8 km | | | |
| Event/ Date | $M_w$ | $D_H$ (km)/ IR (km) | $D_{H'}$ (km)/ $IR_N$ (km) |
| Bhutan Earthquake/ 25th March 2003 | 5.5 | 47.1/ 43 | 49.8/ 41 |
| Jaisalmer Earthquake/ 9th April 2009 | 5.1 | 44.3/ 40 | 49.8/ 34 |
| Khangah Dogran Earthquake/ 19th October 2000 | 4.9 | 47.7/ 26 | 49.8/ 25 |
| Sikkim Earthquake/ 17th June 2018 | 4.4 | 49.8/ NA | 49.8/ 17 |


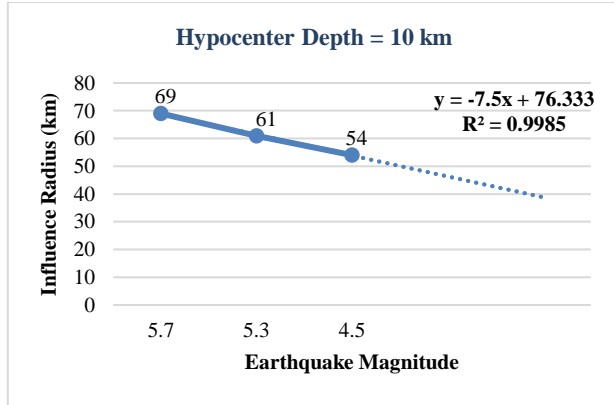
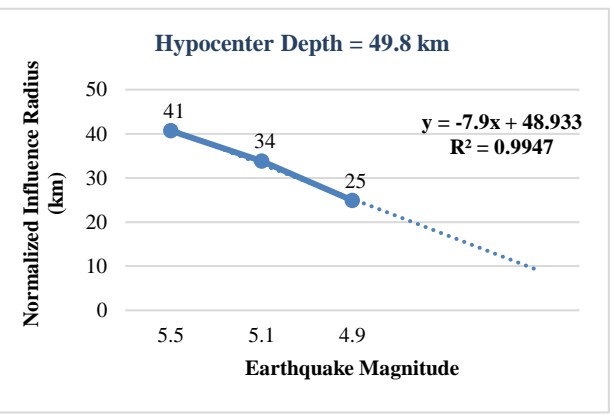



**Figure 14: Relation between IR and Mw of earthquakes triggered at hypocenter depth of 10 km and relation between IRN and Mw of earthquakes triggered close to hypocenter depth of 49.8 km.**

Figure 15A shows the impact of the 2017 Sikkim earthquake (Mw 4.2) with a hypocenter depth of 10 km in terms of the

coherence difference between pre-and co-seismic pairs. A significant reduction in co-seismic coherence has not been observed within the influence area (46 km). However, this is expected as an earthquake event of a lower magnitude (Mw<4.5) does not deliver a substantial impact on the Earth's surface (Lohman and Simons, 2005; Funning and Garcia, 2019). The vertical displacement lies in the range of 0 to -19 mm, indicating subsidence, much less than the earthquake events of higher magnitudes (Mw≥4.5) discussed in the previous sections. The earthquake-induced vertical displacements do not propagate to the glacial

bodies which are clustered at the influence area's northern region. Fig. 16 shows the coherence difference map of the 2018 Sikkim earthquake (Mw 4.4) triggered at a hypocenter depth of 49.8 km. The earthquake results in negligible impact within the influence area of 17 km. Where the epicenter of both the earthquake events of 2017 and 2018 lie within Sikkim at a distance of 31.7 km from each other, the 2017 earthquake (Mw 4.2) shows some subsidence, whereas vertical deformation has not been captured for the 2018 earthquake. The earthquake event that occurred in 2017 is comparatively shallow. InSAR-based

detectable surficial displacement due to a deeper earthquake is smaller than a shallow earthquake of the comparatively same magnitude. To better understand the impact of moderate earthquake events of the same magnitude but different hypocenter depths on glacial bodies, two different earthquake events in the Nepal region have been investigated in the next section.

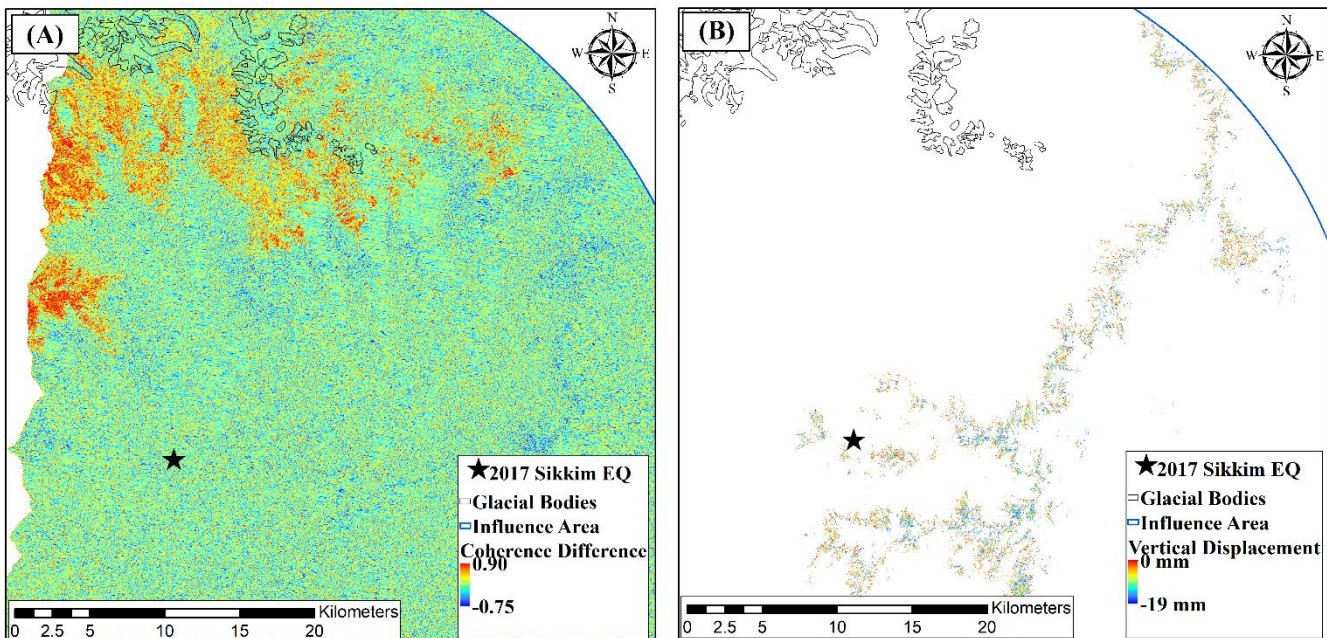

**Figure 15: Map showing (A) difference between pre- and co-seismic coherence and (B) vertical displacement (mm) within influence**
**area derived from unwrapped phase interferogram for 2017 Sikkim earthquake (Mw 4.2).**





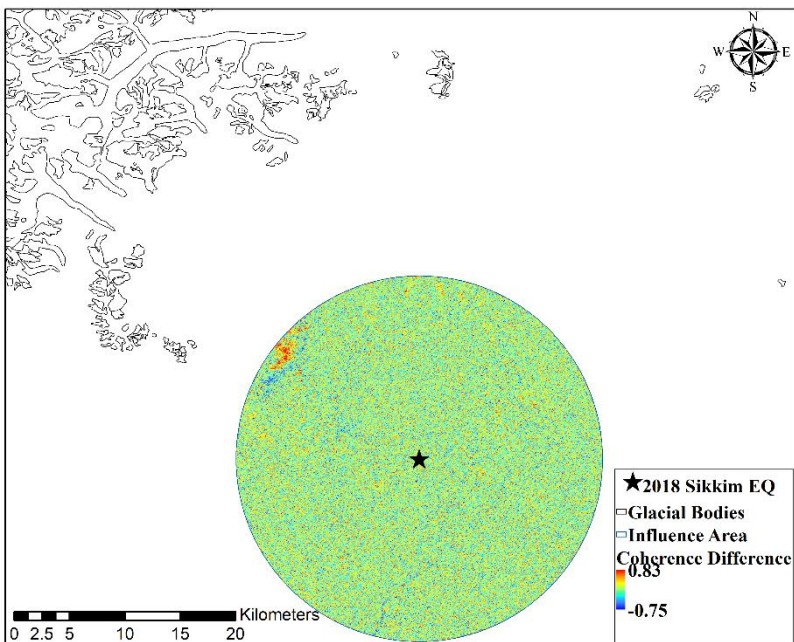

**Figure 16: Map showing difference between pre- and co-seismic coherence for 2018 Sikkim earthquake (Mw 4.4).**

**4.5 2020 and 2021 Nepal Earthquakes, Mw 4.1**

Comparative study on the impact of earthquakes on glacial bodies is a challenging task as no two earthquake events of similar
magnitude get triggered at the same location. Even if two seismic events of similar magnitude occur in the proximity of glacial
bodies, hypocenter depth and distance from the glacial bodies govern the magnitude of vertical deformation. This section
attempts to understand the vertical displacement within glacial bodies due to two different earthquake events of Mw 4.1
triggered at an almost similar distance from the glacial bodies with different hypocenter depths. Despite having similar
magnitudes, the influence radius of both earthquakes is different due to varying hypocenter depths. The influence radius of the
2020 Nepal earthquake triggered at a hypocenter depth of 10 km is computed as 39 km (Table 3). For the 2021 Nepal
earthquake triggered at a hypocenter depth of 35 Km, previous records of earthquake events and their respective shake maps
have been utilized to derive the normalized influence radius of 34 km, as shown in Table 4. The linear relationship between
the normalized influence radius of earthquakes and their magnitudes is represented in Fig. 17.

**Table 4: Earthquakes and their respective magnitudes and influence radii close to hypocenter depth of 35 km**

| Earthquakes with varying hypocenter depth close to 35 km | | | |
|---|---|---|---|
| **Event/ Date** | $M_w$ | $D_H$ (km)/ IR (km) | $D_{H'}$ (km)/ $IR_N$ (km) |
| Dhekiajuli Earthquake/ 28th April 2021 | 6.0 | 34/ 71 | 35/ 70 |
| Haripur Earthquake/ 10th October 2010 | 5.2 | 33.2/ 60 | 35/ 57 |
| Bageshwar Earthquake/ 1st May 2010 | 4.5 | 35.5/ 45 | 35/ 46 |





| Nepal Earthquake/ 11th January 2021 | 4.1 | 35/ NA | 35/ 34 |
|---|---|---|---|


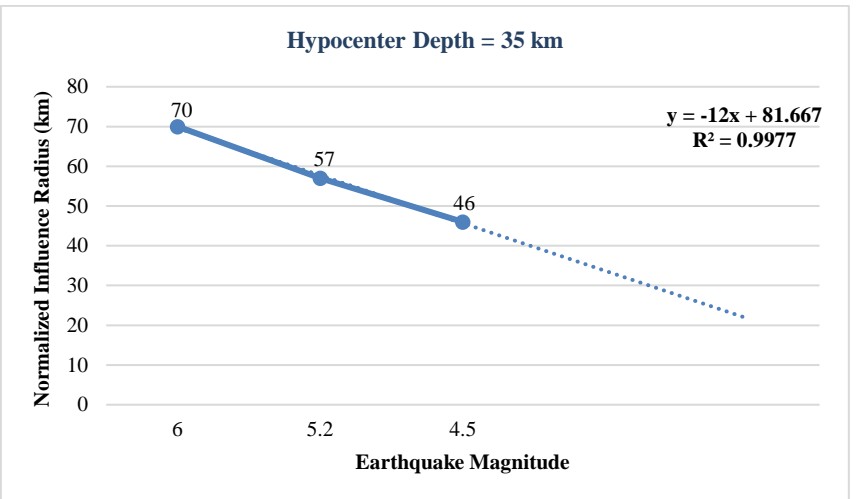

**Figure 17: Relation between IRN and Mw of earthquakes triggered close to hypocenter depth of 35 km.**

The 2020 Nepal earthquake resulted in vertical displacement within the influence area from 0 to -16 mm, indicating surficial sinking (Fig. 18B). The mean displacement is -10 mm. Within the glacial bodies, the mean vertical sinking is 7.7 mm. On the

other hand, the 2021 earthquake resulted in the vertical displacement in the range of 0 to -12 mm with the mean displacement of -5.4 mm, as shown in Fig. 19B, C and D. The earthquake events are triggered at a distance of 33 km from each in the winter of 2020-2021 (within 64 days). A common circular study zone, considering the distance between the two event epicenters as the diameter, is demarcated to capture vertical displacement to two earthquake events of same magnitude but different hypocenter depths positioned at same distance from common glacial bodies. The 2020 earthquake at a comparatively shallow

depth shows much more glacial region with sinking than the 2021 earthquake (Fig. 20). As energy dissipation is more for an earthquake of greater hypocenter depth, shallow earthquakes deliver more impact on the surface that can be clearly seen from the above scenario. Fig. 18A and Fig. 19A show the coherence difference map of the 2020 and 2021 earthquakes, respectively. Significant changes in coherence have not been observed in either case. Even though some locations show reduced co-seismic coherence values, this can be attributed to changes due to other factors. As vegetation cover surrounds the region, as shown in

the False Color Composite (FCC) map (Fig. 21), the coherence difference can be attributed to changes in vegetation cover and not certainly due to earthquakes.



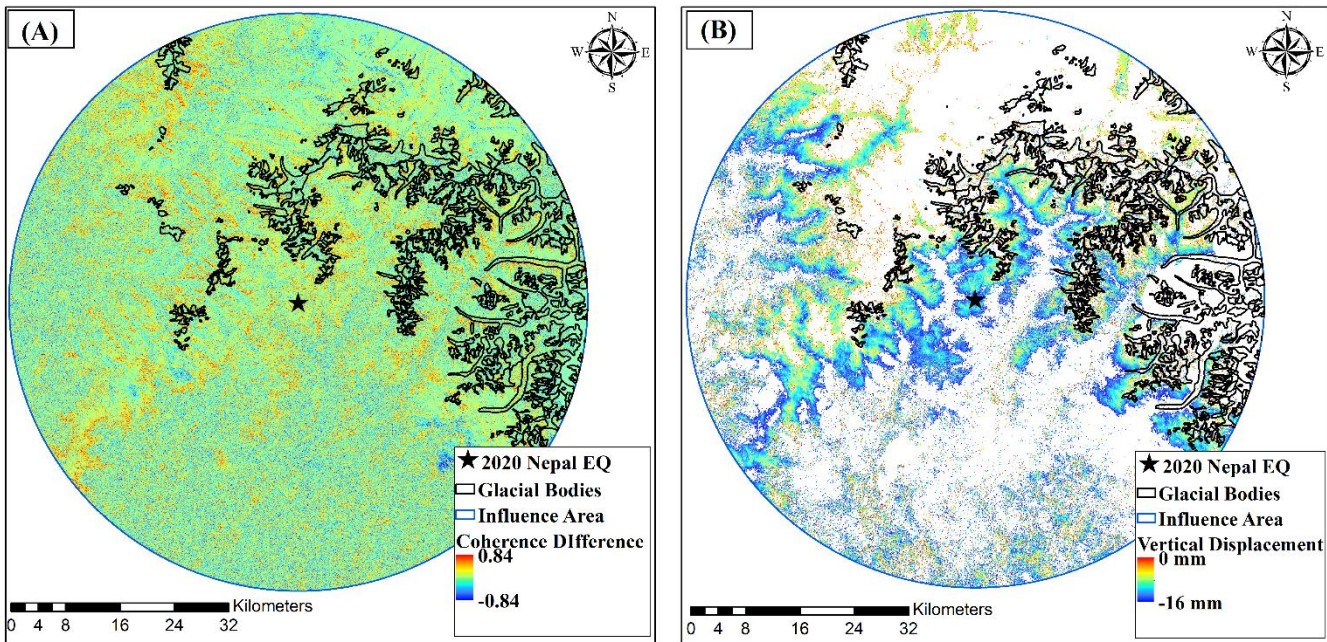

**Figure 18: Map showing (A) difference between pre- and co-seismic coherence and (B) vertical displacement (mm) within influence area derived from unwrapped phase interferogram for 2020 Nepal earthquake (Mw 4.1).**

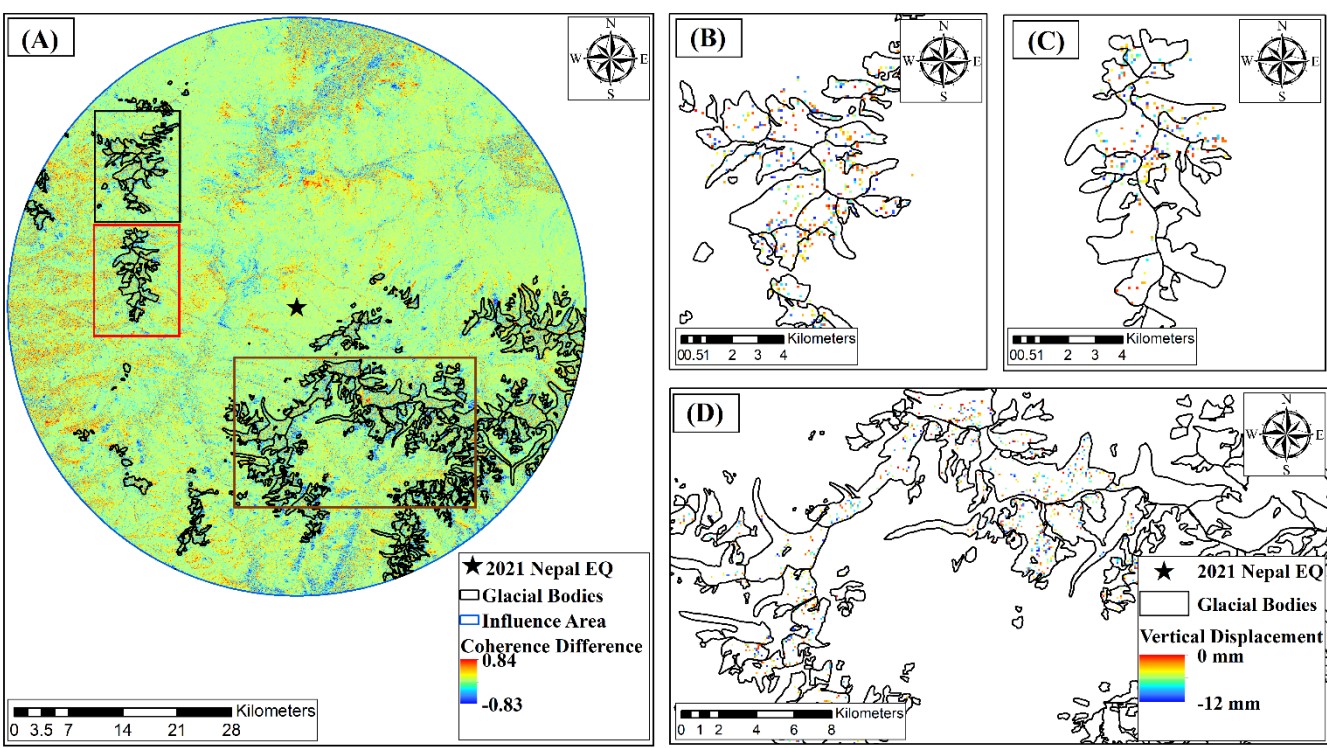




**Figure 19: Map showing (A) difference between pre- and co-seismic coherence, (B), (C) and (D) vertical displacement (mm) within influence area derived from unwrapped phase interferogram for 2021 Nepal earthquake (Mw 4.1).**

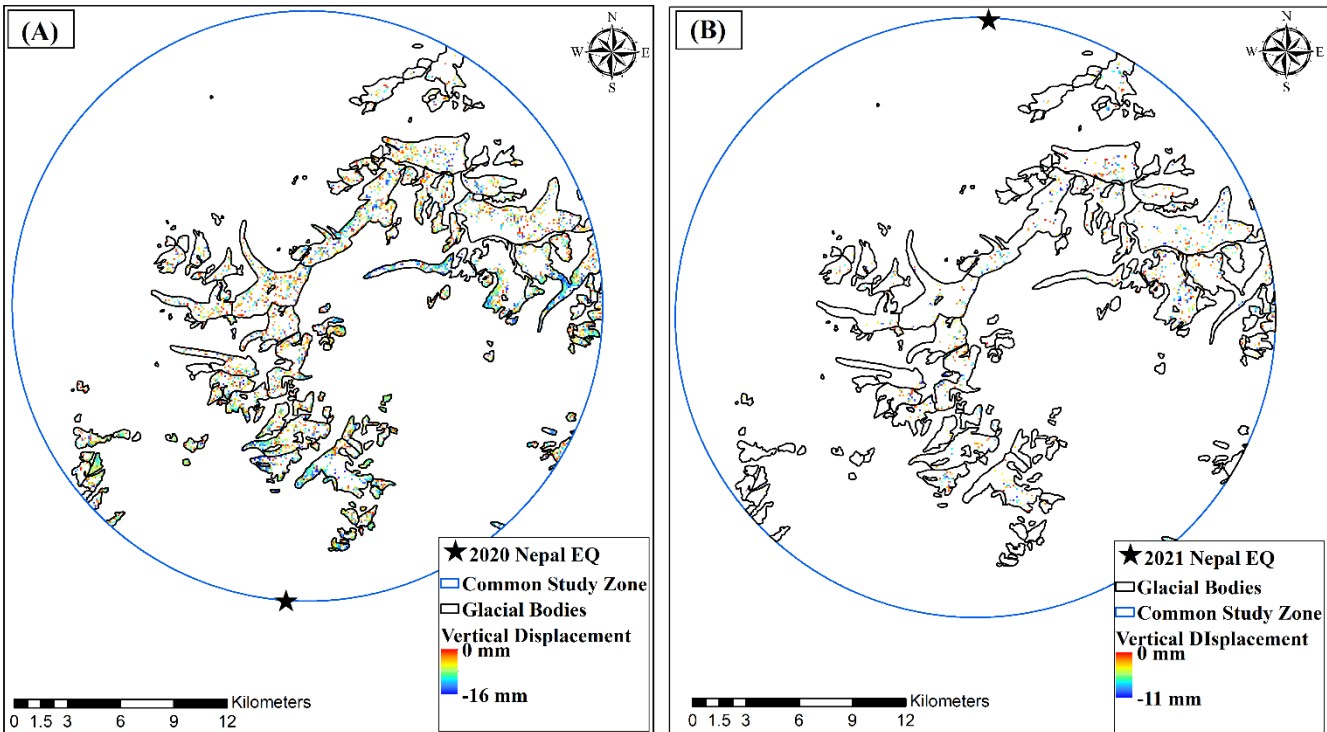

**Figure 20: Map showing vertical displacement (mm) within glacial bodies of common study zone for (A) 2020 Nepal earthquake**
**(Mw 4.1) and (B) 2021 Nepal earthquake (Mw 4.1).**





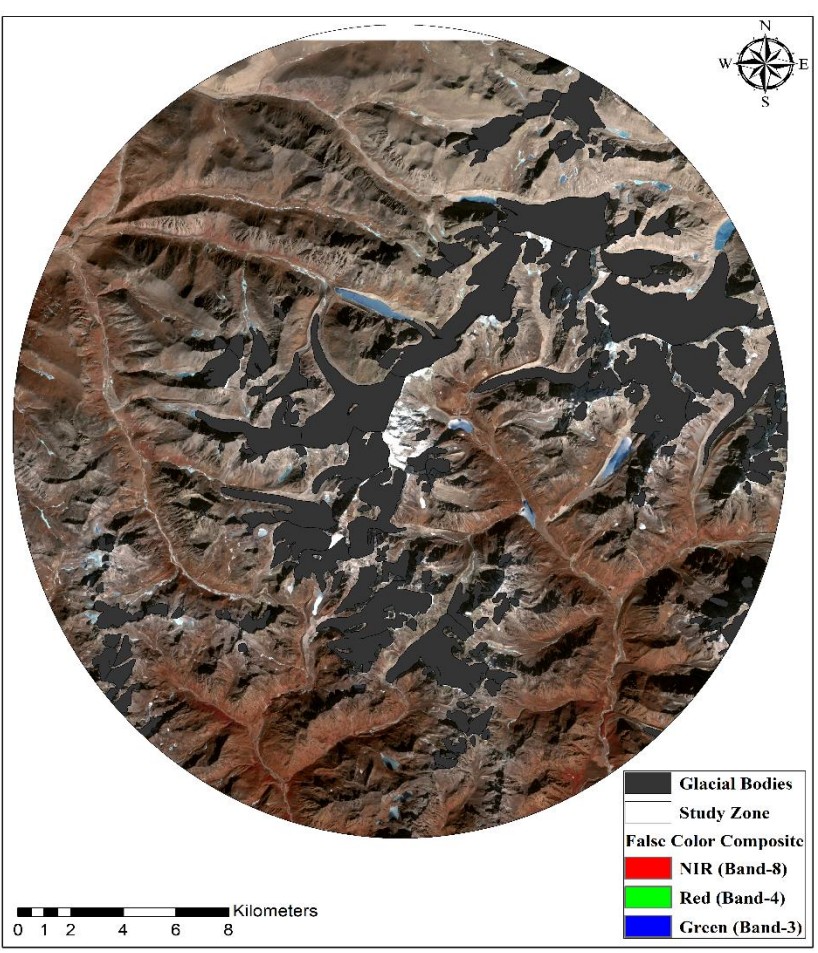

**Figure 21: False Color Composite (FCC) map of the common study zone in Nepal.**

After investigating several earthquake events in the glaciated regions of Himalayan ranges, it can be observed that DInSAR acts as an effective tool in investigating earthquake impacts in areas where other conventional techniques are tough to implement. Where both magnitude and hypocenter depth play a paramount role in the detectability of an earthquake through C-band radar interferometry, this paper attempts to develop an understanding towards seismic displacement of high-altitude Himalayan glaciated regions from western to the eastern ranges considering various categories of earthquakes. Where shake maps can be very useful in determining the influence radius of an earthquake of higher magnitudes (Mw≥4.5), normalized influence radii can be derived for lower magnitude earthquakes for which shake maps are unavailable. Moreover, every earthquake is unique in terms of its impact on the Earth's surface. Where some of the earthquake events such as the 2020 Tibet earthquake, the 2017 Sikkim earthquake, as well as the 2021 and 2021 Nepal earthquakes resulted in negative vertical displacement or subsidence, some of the seismic events such as the 2017 Thang earthquake, 2020 Leh earthquake and 2021 Joshirmath earthquake resulted in both uplift and subsidence.



As observed from multiple earthquake events, the seismic impact on solid earth regions and glacial bodies shows significant
variation in terms of seismic displacements. The vertical displacement of a glacial mass is much higher than the solid earth for
a particular earthquake, which suggests seismic amplification due to the presence of metamorphosed glacial ice over the ground
surface. Moreover, increased vertical displacement of a glacial patch away from epicenter than the glacial patch close to the
epicenter, in the case of the 2017 Thang earthquake, suggests that site-specific seismic amplification properties of glacial
bodies plays a role in the associated displacement. However, it is to be noted that glacial bodies are superimposed on top of
the solid earth. Therefore, the vertical displacements identified via DInSAR may not be solely due to the glacial material alone
but the composite displacement of both glacial body and the solid earth beneath. This cannot be segregated using satellite-
based techniques.

**5 Conclusion**

DInSAR has recently emerged as a powerful tool for studying various natural hazards in the Himalayan region. Especially in
the case of earthquakes, where every earthquake event is unique in terms of its source, magnitude, and impact. Therefore, we
studied seismic events of varying magnitudes and hypocenter depths triggered in the proximity of different glaciated terrains
of the Himalayan region using Sentinel-1A/B C-band DInSAR. The finding of this study are as follows:

• The 2020 Tibet earthquake (Mw 5.7) at a hypocenter depth of 10 km results in large vertical displacements within its
influence area (69 km). A maximum subsidence of 168 mm is observed near the epicenter, with a mean displacement of -47.3
mm. The mean vertical displacement of glacial bodies is found to be -38.9 mm. Reduced co-seismic coherence is observed
around some glaciers. However, such changes are insignificant as the glacial bodies are much away from the earthquake
epicenter.

• In the case of the 2020 Leh earthquake (Mw 5.3), vertical displacement lies in the range of 31 mm and -33 mm, with
a mean displacement of -2.4 mm within the influence area of 61 km. The glacial bodies lying close to the epicenter show
positive displacement, or uplift, that shifts towards subsidence as we move away from the epicenter. The mean glacial
displacement is found to be -2.5 mm. Comparison of glacial and solid earth patches separately, at equal radial distances of 41
km from the epicenter, reveals that the glacial region shows greater vertical displacement (up to -78 mm) than the solid earth.

• Similar to the 2020 Leh earthquake, the 2017 Thang earthquake (Mw 5.2) results in vertical surface displacement that
varies from 51 mm to -39 mm, indicating both uplift and subsidence. The mean displacement of the Earth's surface and glacial
bodies within the influence area (56 km) are -2.8 mm and -3.36 mm, respectively. Comparison of glacial and solid earth
patches, unwrapped separately, followed by vertical displacement, shows much greater values of seismic displacement for
glacial patches that varies up to -173 mm and -576 mm. This exceeds the range of displacements for the solid earth surface by
a large offset, considering either small patches or the entire influence area of the earthquake. Moreover, glacial patch farther
from the epicenter but surrounded by larger glacial bodies show larger values of vertical displacement than the glacial patch
closer to the epicenter, indicating site-specific seismic amplification properties of glaciers.



• The 2021 Joshimath earthquake (Mw 4.5) leads to vertical displacement within its influence area of 54 km that ranges between 45 and -52 mm. The mean displacement of the ground surface and glacial bodies are 0.64 mm and -2.2 mm. From the coherence difference map of the area, a reduction in co-seismic coherence is found around the glacial bodies along the northeastern region of the influence area where surficial uplift is observed. Similar to the 2020 Leh and 2017 Thang

earthquakes, glacial and ground patches considered separately in case of 2021 Joshimath earthquake shows vertical displacement up to -94 mm and -51 mm respectively.

• The 2017 Sikkim earthquake (Mw 4.2), sourced at a depth of 10 km, shows displacement up to -19 mm within the influence radius of 46 km. Due to the lack of shake maps for earthquakes with Mw≤4.5, the influence area is computed using the linear relationship between influence radii and magnitudes of different earthquakes at a hypocenter depth of 10 km. The

2018 Sikkim earthquake (Mw 4.4) at a hypocenter depth of 49.8 km shows no vertical displacement within the normalized influence area of 17 km.

• The 2020 Nepal earthquake (Mw 4.1) shows vertical displacement between 0 and -16 mm with mean subsidence of 6.2 mm within the computed influence radius of 39 km. The mean subsidence of glacial bodies is derived as 7.7 mm. Whereas the 2021 Nepal earthquake (Mw 4.1) shows vertical displacement in the range of 0 to -12 mm, having a mean subsidence of

5.4 mm within the normalized influence radius of 34 km. Both earthquake events are triggered at a distance of 33 km from each other within 64 days. Studying two different earthquake events of similar magnitude located at almost the same distance from the glacial bodies but having different hypocenter depths show that a shallower earthquake has more impact on glacial bodies in terms of vertical displacement than a deeper one.

Thus, earthquake events around Himalayan glaciers can be well studied with the help of C-band SAR interferometry, where

other geodetic and photogrammetric techniques are hard to implement. Moreover, the continuous data acquisition of orbiting radars at a certain temporal offset makes them effective in studying earthquakes which are very unpredictable in terms of their triggering location and magnitude.

Glacial bodies are a dense and thick envelope of metamorphosed ice that may not reproduce similar deformation as the solid earth surface and may result in site-specific amplification of seismicity. This suggests that because glaciers are different in

terms of their composition and behavior, their response to seismic tremors is also different. As glaciers have viscoelastic properties, earthquakes, in terms of both the associated deformation and response to seismic waves, may impact a glacial mass differently than solid bedrock. Future research into the dynamics behind seismic deformation and amplification of glacial bodies should incorporate extensive modeling of that interaction.

As observed from analyzing multiple earthquake events, glacial mass studied separately using DInSAR reveal site-specific

seismic amplification as compared to the solid earth surface. However, glacial bodies are thick chunks of metamorphosed ice lying over the ground underneath. Therefore, the vertical displacement of glacial bodies registered using DInSAR is likely the composite displacement of both glacier and the solid earth surface at the base of the glacier. To develop a better understanding of the impact of seismicity on glaciers, will require additional studies into the material and dynamic properties of glacial bodies.



This study indicates that the effect of earthquake events on glaciers can be much more significant than their impact on the surrounding terrain. As earthquakes can also induce other disasters which includes glacial surging, ice avalanches, landslides and failure of moraine-/ice-dammed glacial lakes, a relatively mild earthquake on the surface can pose substantial risks to the surrounding glacier masses. Therefore, studying the impact of earthquakes on glaciers and further connecting it with glacial geomorphology, geological and lithological settings can assist in identifying at-risk areas and contribute to risk mitigation planning in the near future.

*Data availability.* *The datasets used in this study are available for download from the web portals mentioned as follows:*

*Sentinel-1 SLC datasets:* *https://search.asf.alaska.edu/#/*

*Sentinel-2 MSI datasets:* *https://scihub.copernicus.eu/dhus/#/home*

*ZTD Maps:* *http://www.gacos.net/*

*Glacial Shape Files:* *https://www.glims.org/RGI/*

*Author contribution.* Sandeep Kumar Mondal: Conceptualization, Formal Analysis, Software, Writing - original draft. Rishikesh Bharti: Conceptualization, Supervision, Writing - review & editing. Kristy F Tiampo: Conceptualization, Writing - review & editing.

*Competing interests.* The authors declare no potential competing interest that could have influenced the study reported in this paper.

*Acknowledgements.* We are grateful to European Space Agency (ESA), Alaska Satellite Facility (ASF), United States Geological Survey (USGS), and Randolph Glacier Inventory (RGI 6.0) supplemental to the Global Land Ice Measurements from Space initiative (GLIMS) for their tremendous efforts to make the data available for the current study.

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
