# Peer review of "Seismic Deformation of Himalayan Glaciers Using Synthetic Aperture Radar Interferometry"

_EGUsphere, 2023_

## Author Comment (AC1)

**RESPONSE TO EDITOR'S/REVIEWER'S COMMENTS**

We would like to thank the editor and reviewers for sparing their valuable time in reading our manuscript and providing valuable comments/suggestions which have helped us to improve earlier version of the manuscript. In view of the comments/suggestions, we have revised our manuscript, hope our manuscript will be acceptable to the Referees and to the Editor. The suggested changes are highlighted (with green colour) in the manuscript.

**Reply to the points raised by the Reviewer (RC1)**

**General comments**

The study presented demonstrates a commendable use of English. There is major room for improvement in the presentation of the figures. Overall, the chosen subject matter is captivating.

Nevertheless, problems of basic methodological elements weaken the study considerably.

For this reason at this point I would propose a major revision.

**Answer:** We are thankful to the reviewer for reading the manuscript and suggesting revision.

**Specific comments**

The authors employ the InSAR technique. The theoretical part of the SAR methods adopted are clearly explained and in detail. However, at first the authors focus on the Tibet 2020 earthquake (Mw5.7), which caused significant deformation detectable by InSAR, as evident from the fringes displayed in the wrapped interferogram in Fig. 3D. However, starting from the second event studied (Leh earthquake, Mw5.3), the magnitudes are too low to be reliably detected by InSAR. Additionally, subsequent minor events with magnitudes ranging from 4.1 to 4.5 and varying depths of 35, 49, or even 74 kilometers, further challenge the feasibility of using InSAR for surface deformation monitoring. Consequently, it becomes apparent why the wrapped interferograms for these events are not presented -except for the Mw5.7 Tibet 2020 earthquake. Thus, the approach utilized to measure surface deformation for the second event (Leh earthquake, Mw5.3, depth: 10 kilometers) could be questionable, and for all subsequent events, it is deemed unsuitable.

**Answer:** In this work, we attempt to capture surface and glacial deformations due to earthquake events of various magnitude and hypocentre depth. The idea is to check the capability of the C-band radar interferometry technique to capture seismic deformation of glacial bodies and ground surface.

It is true that the detectability of reliable deformation reduces with decrease in earthquake magnitude and increase in hypocentre depth. Therefore, in order to capture the reliable seismic deformation, we have considered deformation of the regions with co-seismic coherence $\geq 0.6$. Our results align with the general understanding of earthquake deformation detectability and also with the results presented by Li et al., 2022, as explained in the manuscript.

In fact, for the earthquake events of lesser magnitudes, the detection of reliable seismic deformation is questionable, as in the case of the 2017 Sikkim Earthquake ($M_w$ 4.2), 2018 Sikkim Earthquake ($M_w$ 4.4) as well as 2020 and 2021 Nepal Earthquakes ($M_w$ 4.1).

The wrapped and unwrapped interferograms for every earthquake event considered in this study is presented in the supplementary file along with the shake maps, isoseismal contours and focal mechanism (available for whichever earthquakes in the USGS Earthquake Catalog and Global Centroid

**RESPONSE TO EDITOR'S/REVIEWER'S COMMENTS**

Moment Tensor (CMT) Catalogue respectively). Those results support these conclusions, but also identify other features, as discussed in the paper.

The authors make assumptions regarding the influence radius of the smaller events, their methodology could potentially be considered to lack sufficient data to ensure reliable results. However, what is more crucial is the absence of estimations for the expected deformation of each event. I consider, that it is imperative for the authors to develop forward surface displacement models, potentially based on formulations such as those of Okada. This missing element is of significant importance. Such models could be constructed using the focal mechanisms of the events or -for more detailed results- on slip distributions, if available. Understanding the anticipated surface deformation based on the event source is paramount, allowing for a comparison with observed patterns to ascertain whether the glaciers are following expected trends or displaying deviation.

**Answer:** We would like to thank the reviewer for the valuable suggestion. As isoseismal contours and shake maps with Modified Mercalli Intensity (MMI) scale are available online for earthquake events with much higher magnitudes, understanding seismic deformations for comparatively moderate and lower magnitude earthquakes become difficult. Similarly, focal mechanisms are available online for higher magnitude earthquakes only. In particular, for the earthquake events which are triggered at high-altitude Himalayan regions where glaciers are present, such information is unavailable. In this context, C-band radar interferometry can act as an effective tool in studying earthquakes which are triggered near remotely located high-altitude glacial bodies.

The USGS shake maps, isoseismal contours and focal mechanisms for 2020 Tibet earthquake (Mw 5.7), 2020 Leh earthquake (Mw 5.3) and 2017 Thang earthquake (Mw 5.2) and shake map for 2021 Joshimath earthquake (Mw 4.5) is presented in the supplementary file.

We first tested the proposed methodology for the 2020 Tibet Earthquake ($M_w$ 5.7), which was studied extensively for surface deformation. Later, we utilized the method for other earthquakes of varying magnitudes and hypocentre depths.

**Technical corrections**

1. Add the wrapped and unwrapped interferograms for all the events.

**Answer:** The wrapped and unwrapped interferograms for all the earthquake events studied in this work are presented in the supplementary material.

2. Add the available shakemaps in the figures, or in the supplementary material.

**Answer:** The shakemaps and isoseismal contours are presented in the supplementary material.

3. Add a global map where you can show the area of interest. Also since you refer to them, add the Indian and Eurasian plate boundaries, Tibetan plateau Himalayan frontal thrust etc. All that is mentioned in the text, would be better to be shown in the figures too.

**Answer:** We thank the reviewer for the suggestion. Figure-1 has been modified with a global map, Indian:Eurasian plate boundaries and tectonic lineaments.

4. According to gacos.net, when using GACOS, all the following papers should be cited:

- Yu, C., Li, Z., Penna, N. T., & Crippa, P. (2018). Generic atmospheric correction model for Interferometric Synthetic Aperture Radar observations. Journal of Geophysical Research: Solid Earth, 123(10), 9202-9222.

**RESPONSE TO EDITOR'S/REVIEWER'S COMMENTS**

- Yu, C., Li, Z., & Penna, N. T. (2018). Interferometric synthetic aperture radar atmospheric correction using a GPS-based iterative tropospheric decomposition model. Remote Sensing of Environment, 204, 109-121.

- Yu, C., Penna, N. T., & Li, Z. (2017). Generation of real-time mode high-resolution water vapor fields from GPS observations. Journal of Geophysical Research: Atmospheres, 122(3), 2008-2025.

**Answer:** Suggested papers have been cited.

5. Add the focal mechanism of all the events in the corresponding figures.

The focal mechanisms for 2020 Tibet earthquake (Mw 5.7), 2020 Leh earthquake (Mw 5.3) and 2017 Thang earthquake (Mw 5.2) are presented in the supplementary file.

6. In section 4.2 around line 245 you mention: "…regions close to the epicenter location show positive vertical displacement for the 2020 Leh earthquake, with negative vertical displacements moving away from the epicenters." This is an example where you should present what you measured and which is the theoretically expected deformation pattern.

[Figure]

**RESPONSE TO EDITOR'S/REVIEWER'S COMMENTS**

**Answer:** We thank the reviewer for the suggestion. In case of the 2020 Leh Earthquake, we observed upliftment near epicentre that later transforms to subsidence as moving away from the epicentre. We have checked for various other earthquakes including the recently occurred 2024 China Earthquake sequence which includes a mainshock (Mw 7.0) at a hypocentre depth of 13 km and two aftershocks (Mw 5.8 and 5.5) at a hypocentre depth of 10 km in the Tien Shan mountains' fold and thrust belt shows similar observation (as shown in the map above). Both the earthquakes have occurred due to thrust faulting as understood from the focal mechanism and beach ball map of 2024 China Earthquake (given below). The focal mechanism and beach ball map for 2020 Leh earthquake is shown in the supplementary file.

| Earthquake | Event ID | Latitude (°) | Longitude (°) | Depth (km) | Nodal Plane-I | | | Nodal Plane-II | | | $M_w$ | Source |
|---|---|---|---|---|---|---|---|---|---|---|---|---|
| | | | | | *Strike-I* | *Dip-I* | *Rake-I* | *Strike-II* | *Dip-II* | *Rake-II* | | |
| | | 41.2628 | 78.6594 | 22 | 108.39 | 55.86 | 144 | 220.51 | 60.96 | 39.94 | 6.7 | NEIC |
| **Mainshock** | 636644404 | 41.294 | 78.594 | 22 | 105 | 53 | 127 | 234 | 50 | 51 | 7.1 | IPGP |
| | | 41.19 | 78.56 | 16.1 | 112 | 60 | 127 | 235 | 46 | 44 | 7 | GCMT |
| **Aftershock** | 641513987 | 41.2023 | 78.7264 | 10 | 91.12 | 38.32 | 97.7 | 261.34 | 52.09 | 83.95 | 5.1 | NEIC |
| | | 41.21 | 78.54 | 23.2 | 98 | 42 | 101 | 262 | 49 | 80 | 5.5 | GCMT |

7. Figure 5A. Why is there such a sharp discontinuity?

**Answer:** This discontinuity is likely the result of phase unwrapping errors associated with the topography. This is a good example of why we masked out those areas with coherence less than 0.6, resulting in Figure 5B, which shows areas of more reliable results.

6. Provide a citation for the SRTM DEM.

**RESPONSE TO EDITOR'S/REVIEWER'S COMMENTS**

**Answer:** The following citation has been included in the manuscript for the SRTM DEM:

Rabus, B., Eineder, M., Roth, A. and Bamler, R., 2003. The shuttle radar topography mission—a new class of digital elevation models acquired by spaceborne radar. ISPRS journal of photogrammetry and remote sensing, 57(4), pp.241-262. https://doi.org/10.1016/S0924-2716(02)00124-7

**FINAL COMMENTS**

I find the arguments of the authors intriguing: the presence of metamorphosed glacial ice over the ground surface leads to seismic amplification and the effects of earthquake events on glaciers can be notably more significant than their impact on the surrounding terrain. But as previously mentioned, the analysis falls short in terms of robustness. The methods adopted in this manuscript are not suitable for most of the part of the study that the authors present.

However, considering that the subject is interesting, I would invite the authors to present an improved version of it, after a major revision.

**Answer:** We agree with the reviewer that the results are intriguing. While it is regrettable that there is not more complete seismic information available in the region, we believe that these first results are of value to the community and that future work will better elucidate the mechanisms and dynamics. We have included this discussion in the conclusion.

---

## Author Comment (AC2)

**RESPONSE TO EDITOR'S/REVIEWER'S COMMENTS**

We would like to thank the editor and reviewers for sparing their valuable time in reading our manuscript and providing valuable comments/suggestions which have helped us to improve earlier version of the manuscript. In view of the comments/suggestions, we have revised our manuscript, hope our manuscript will be acceptable to the Referees and to the Editor. The suggested changes are highlighted (with green colour) in the manuscript.

**Reply to the points raised by the Reviewer (RC2)**

In this study the authors attempt to correlate deformation measured using the InSAR technique from minor earthquakes with magnitudes ranging between Mw 4.1 and 5.7 with deformation at nearby glaciers. While the idea in principle is interesting and worthy of investigation, I am not persuaded by the arguments presented in this study of the correlation between earthquakes and glacial deformation.

**Answer:** We are thankful to the reviewer for reading the manuscript and suggesting revision.

**Major:**

- Fundamentally I am unconvinced by your arguments that these small (Mw<5.7) earthquakes have resulted in deformation at glaciers. You've shown interferograms of several earthquakes and unconvincing maps of deformation at glaciers. However, the causality is lacking. One convincing way to do this would be to generate a deformation time series at the glacier and how that there is an acceleration following the earthquake.

**Answer:** In this study, we attempted to capture reliable seismic deformation due to earthquake tremors of different magnitudes and hypocentre depths within glacial bodies as well as ground surface. In support of our analysis, we have now presented the wrapped and unwrapped interferograms along with the shake maps, isoseismal contours and focal mechanism in the supplementary file.

- It's not clear to me why you have only selected earthquakes between magnitudes 4.1 and 5.7. Why have you missed out the potential impacts from larger events, for example the 2016 Nepal earthquake?

**Answer:** We checked for the 2015 Nepal earthquake ($M_w$ 7.8). However, we could not generate interferogram using DInSAR. We also studied recently occurred 2023 Nepal earthquake sequence ($M_w$ 5.7 mainshock and $M_w$ 5.3 aftershock) and 2024 China earthquake sequence ($M_w$ 7.0 mainshock and $M_w$ 5.8 & 5.5 aftershocks).

**RESPONSE TO EDITOR'S/REVIEWER'S COMMENTS**

[Figure]

**Figure-** Reliable vertical ground deformations due to 2023 Nepal Earthquake Sequence

[Figure]

**Figure-** Reliable vertical ground deformations due to 2023 Nepal Earthquake Sequence

We could observe maximum atmospherically-corrected ground deformation of -79 mm as shown below for 2023 Nepal earthquake and 239mm for 2024 China Earthquake. However, since major glacial bodies were not lying within the influence area, we could not capture glacial deformations. The work based on

**RESPONSE TO EDITOR'S/REVIEWER'S COMMENTS**

2023 Nepal earthquake sequence has been presented in the EGU General Assembly 2024 (***Link:*** https://doi.org/10.5194/egusphere-egu24-17792) and 2024 Chain earthquake sequence is submitted in IEEE InGARSS India Geoscience and Remote Sensing Symposium 2024.

- You are missing a step in your assumption that the InSAR data provides vertical deformation (e.g. Figure 2, Line 161). It doesn't, it gives you deformation in the Line-of-Sight direction. You can assume that all your deformation is vertical, which is not great but fine. But you need to clearly state this assumption and discuss the impacts of it in your discussion.

**Answer:** We are thankful to the reviewer for the suggestion. We have mentioned in Line 146 and 147 as "Division by the cosine of the incidence angle converts the line-of-sight (LOS) displacement to vertical displacement."

So, the displacement represented in the manuscript are vertical displacements and not LOS displacement.

- You don't say anything about how you do the spatial reference selection for the InSAR images. This is particularly important when you are looking at differences between pre- and coseismic image pairs. This also becomes extremely problematic when you unwrap areas separately as you say you do in Lines 258-261. Did you pick the same point for the both the pre- and co-earthquake pairs? If not, how do you know the differences you see are not due to reference selection?

**Answer:** The temporal baseline for the selection of Sentinel-1 SLC datasets is 12 days. Therefore, if ascending pass is considered, the pre- and co-seismic data pairs have a temporal difference of 12 days. The same is for descending pass datasets. The details are shown in Table-2.

Yes, we have picked the same region for pre- and co-earthquake pairs when we are looking into the deformation of glacial subset regions.

Unwrapping the DInSAR phase information for the entire earthquake-affected region can lead to unwrapping errors. Therefore, we separately examined the vertical displacement caused by seismic events on glaciers and the surrounding ground surface as subset regions. This approach helped avoid unwrapping errors when dealing with a larger area.

- To me looks like there are major unwrapping errors in Figures 5/11. The jump between and red (uplift) and subsidence (blue) is far too sharp. Please can you confirm and show that this is not the case. I also can't see the earthquake in the interferograms. For all earthquake figures can you plot the wrapped image like you did for Fig 4, please also plot the epicentre location and the USGS contours you use to delineate the influence area.

**Answer:** We checked for the 2020 Leh earthquake (Mw 5.3) and 2021 Joshimath earthquake (Mw 4.5) by performing the unwrapping in both SNAP as well as separately in SNAPHU through command prompt. We obtained similar results. In order to better understand the glacial and ground deformations, we then subset the glacial and ground areas and then performed unwrapping as shown in Figure-7 and Figure-13 respectively. The earthquake (as fringes in the phase information) cannot be seen in the interferogram as the magnitude is less than 5.5. We observed this for other earthquakes as well such as 2016 Kumamoto Earthquake (mainshock of $M_w$ 7.0), 2023 Nepal Earthquake ($M_w$ 5.7) and recently occurred China Earthquake (mainshock of $M_w$ 7.0) but for earthquakes less than $M_w <5.5$, concentric fringes are not captured.

**RESPONSE TO EDITOR'S/REVIEWER'S COMMENTS**

1. Mondal, S.K. and Bharti, R., 2023, December. Seismicity Declustering and Impact Assessment of Kumamoto Region for 2016 Earthquake Sequence. In 2023 IEEE India Geoscience and Remote Sensing Symposium (InGARSS) (pp. 1-4). IEEE, 10.1109/InGARSS59135.2023.10490367.

2. Mondal, S. K., Bharti, R., and Tiampo, K.: Seismic Deformations due to 2023 Nepal Earthquake Sequence using Satellite Remote Sensing Techniques, EGU General Assembly 2024, Vienna, Austria, 14–19 Apr 2024, EGU24-17792, https://doi.org/10.5194/egusphere-egu24-17792, 2024.

The wrapped and unwrapped outputs, USGS shake maps, isoseismal contours and focal mechanism solutions for different earthquakes are provided in the supplementary file.

- I'm not entirely sure I agree with your influence radius argument. Mostly because I'm not clear what you mean by the influence radius. The shakemap contours are the shaking intensity contours, not the deformation radius. It isn't clear how you've done your calculations in section 4.4. I'm not entirely sure what the 4$^{th}$ column in Tables 3 and 4 represents. How reasonable is it to assume a linear relationship between IR and Mw and hypocentral depth, when we know that in terms of energy release Mw does not scale linearly? Please provide extra clarity on this point.

**Answer:** We have tried to define influence radius for the purpose of delineating glacial bodies that are to be studied for seismic deformations. In order to do so, we first observed the isoseismal contours and checked for the Modified Mercalli Intensity (MMI) chart. The details of the shake maps, isoseismal contours and MMI chart are provided in the supplementary file.

The Modified Mercalli Inetnsity (MMI)-V Isoseismal contour has been considered as threshold to obtain the influence radius of an earthquake in the presented study. The reason for considering MMI Intensity-V is the damage (very light) associated with it. Below MMI Intensity-V, there is no damage as observed from the MMI chart and the PGA and PGV values fall significantly with very light shaking.

| SHAKING | Not felt | Weak | Light | Moderate | Strong | Very strong | Severe | Violent | Extreme |
|---------|----------|------|-------|----------|--------|-------------|--------|---------|---------|
| DAMAGE | None | None | None | Very light | Light | Moderate | Moderate/heavy | Heavy | Very heavy |
| PGA(%g) | <0.0464 | 0.297 | 2.76 | 6.2 | 11.5 | 21.5 | 40.1 | 74.7 | >139 |
| PGV(cm/s) | <0.0215 | 0.135 | 1.41 | 4.65 | 9.64 | 20 | 41.4 | 85.8 | >178 |
| INTENSITY | I | II-III | IV | V | VI | VII | VIII | IX | X+ |

Scale based on Worden et al. (2012)    Version 1: Processed 2022-01-25T17:06:17Z
△ Seismic Instrument  ○ Reported Intensity    ★ Epicenter

*MMI Chart for 2020 Tibet Earthquake ($M_w$ 5.7)*

| SHAKING | Not felt | Weak | Light | Moderate | Strong | Very strong | Severe | Violent | Extreme |
|---------|----------|------|-------|----------|--------|-------------|--------|---------|---------|
| DAMAGE | none | none | none | Very light | Light | Moderate | Moderate/Heavy | Heavy | Very Heavy |
| INTENSITY | I | II-III | IV | V | VI | VII | VIII | IX | X+ |

Processed: Fri Sep 25 21:22:14 2020 vmdyfi1

*MMI Chart for 2020 Leh Earthquake ($M_w$ 5.3)*

In case of 2020 Tibet, 2020 Leh earthquake and 2021 Joshimath earthquake, we used MMI information to define the influence radius. All these earthquakes were triggered at same hypocentre depth of 10 Km and the influence radius of these earthquakes differs linearly with magnitude as shown in Figure-14

**RESPONSE TO EDITOR'S/REVIEWER'S COMMENTS**

(left graph). Thus, the influence radius of the 2017 Sikkim earthquake triggered at hypocentre depth of 10 Km is calculated using the linear relationship shown in Figure-14 (left graph).

In case of the 2018 Sikkim earthquake, the scenario is slightly different. The earthquake is triggered at a hypocenter depth of 49.8 Km. From past records, earthquakes at exactly the same hypocenter depth have not occurred in the Indian subcontinent. Thus, earthquakes of different magnitudes and hypocenter depths close to 49.8 Km have been considered, and their influence radii have been determined from USGS shake maps, as shown in Table 3. However, their relation ($IR$ $vs$ $M_w$) is not linear. Therefore, the normalized influence radius ($IR_N$) for these earthquake events at a hypocenter depth of 49.8 km has been computed using Eq. (7).

$$IR_N = \frac{D_H}{D_{H'}} \times IR$$

Here, $D_{H'}$ represents the hypocenter depth of 2018 Sikkim earthquake for which $IR_N$ of each earthquake is calculated. This develops a linear relationship between $IR$ and $M_w$ as shown in Figure-14 (right graph). Thus, the normalized influence radius of the 2018 Sikkim earthquake is computed as 17 km.

**Minor:**

The introduction is largely irrelevant. It can be improved by removing all the sections giving a background to the techniques and instead focusing on the main aims/purpose of your study. I would say the same for your Data section, you don't need to go into so much unnecessary detail about the system/data specifics of the Sentinel-1/2 satellites.

**Answer:** Suggested modifications have been made.

Line 17: Shakemaps from where? Did you calculate these yourself?

**Answer:** The shakemaps used in this study are obtained from the USGS earthquake catalog. The term "shake maps" have been updated as "USGS shake maps" in Line 17.

Line 18-19: I think this statement needs more information/detail.

**Answer:** We have addressed this in lines 307-327 in the manuscript; please let us know if it requires additional explanation.

Line 43: …assessment is a crucial…

**Answer:** Correction has been made in the manuscript.

Line 54:  … with the help of Synthetic Aperture Radar (SAR)…

**Answer:** Correction has been made in the manuscript.

Line 104: …resolution of approximately 5m×20m…

**Answer:** Correction has been made in the manuscript.

Sections 3.3 and 3.4: Much of the text in these sections is unnecessary. Most of the processing steps and background are largely known and now routine and can be cited away. Please just state what you did and focus on anything you've done differently to the established norm. For

**RESPONSE TO EDITOR'S/REVIEWER'S COMMENTS**

example all the text explaining the background of coherence is not needed. Also, please state which software you used to do this processing, e.g. SNAP, ISCE, GAMMA etc.?

**Answer:** Suggested modifications have been made in Sections 3.3 and 3.4. SNAP and SNAPHU has been used for processing as mentioned in Line 159 and 160.

Line 130: Please reference in the text the section in the paper where you do this (section 4.4).

**Answer:** (refer section 3.4) has been included at the end of Line 130.

Line 166: What resolution? Also, how does the ZTD delay maps help correct unwrapping errors?

**Answer:** The ZTD maps were obtained from GACOS at 3 arcsec (~90m) ground resolution.

Atmospheric effects are a significant source of errors in repeat-pass Interferometric Synthetic Aperture Radar (InSAR) measurements, potentially obscuring actual displacements caused by earthquakes (Yu et al., 2018). Correcting interferograms for atmospheric influences reduces phase delays caused by atmospheric variations between two image acquisitions (Vaka et al., 2019).

- Yu, C., Li, Z. and Penna, N.T., 2018. Interferometric synthetic aperture radar atmospheric correction using a GPS-based iterative tropospheric decomposition model. Remote Sensing of Environment, 204, pp.109-121.

- Vaka, D.S., Rao, Y.S. and Bhattacharya, A., 2021. Surface displacements of the 12 November 2017 Iran–Iraq earthquake derived using SAR interferometry. Geocarto International, 36(6), pp.660-675.

Line 173: I'm not sure this makes sense. Interferometry is not the complex coherence…

**Answer:** We thank the reviewer for suggesting the correction. The correction has been made in the manuscript.

Line 224: Please state which isoseismal contour threshold you used. Can you also plot these on Figure 3?

[Figure]

**RESPONSE TO EDITOR'S/REVIEWER'S COMMENTS**

| SHAKING | Not felt | Weak | Light | Moderate | Strong | Very strong | Severe | Violent | Extreme |
|---|---|---|---|---|---|---|---|---|---|
| DAMAGE | None | None | None | Very light | Light | Moderate | Moderate/heavy | Heavy | Very heavy |
| PGA(%g) | <0.0464 | 0.297 | 2.76 | 6.2 | 11.5 | 21.5 | 40.1 | 74.7 | >139 |
| PGV(cm/s) | <0.0215 | 0.135 | 1.41 | 4.65 | 9.64 | 20 | 41.4 | 85.8 | >178 |
| INTENSITY | I | II-III | IV | V | VI | VII | VIII | IX | X+ |

Scale based on Worden et al. (2012)  Version 1: Processed 2022-01-25T17:06:17Z
△ Seismic Instrument  ○ Reported Intensity  ★ Epicenter

**Answer:** The Modified Mercalli Inetnsity (MMI)-V Isoseismal contour has been considered as threshold to obtain the influence radius of an earthquake in the presented study as shown in the above figures. The reason for considering MMI Intensity-V is the damage (very light) associated with it. Below MMI Intensity-V, there is no damage as observed from the MMI chart and the PGA and PGV values fall significantly with very light shaking as shown above.

The shake maps and isoseismal contours (from USGS Earthquake catalog) for the earthquake events investigated in the presented study are shown in the supplementary file.

Line 226: Are you sure your subsidence is 47.3mm? The blue colours in Figure 3 correspond the subsidence ~150mm!

**Answer:** 47.3mm is the mean subsidence within the influence area whereas in Figure-3, 168mm is the maximum subsidence close to epicentre.

Figure 4: In panel A, please label the boxes that represent panels B and C.: You say that you can see a drop in coherence. I can't see this. Could you show a zoom in of the coherence change? Please plot on the map the earthquake location. Can you do this for all panelled figures (e.g. Fig 12) as it is confusing to see which panel corresponds to which geographic location.

**Answer:** We thank the reviewer for suggestion. The caption of Figure-4 and Figure-12 has been modified mentioning the black and brown regions used in the map. The earthquake locations (epicentre) has been shown in every map. However, where some map panels show smaller regions of glaciers and ground surface, it is not possible to show the epicentre as then the analysis results (deformations and coherence difference) will be within too small area to interpret.

In the map shown below, the glacial locations have been zoomed to show reduced coherence (in blue patches) around glacial bodies. Within the map shown in the manuscript and also the map shown below, regions shown in blue represent reduced co-seismic coherence.

**RESPONSE TO EDITOR'S/REVIEWER'S COMMENTS**

[Figure]

Line 254: I'm not sure I understand what you're saying here. Fig 6B is the unwrapped vertical deformation. If it is difficult, how did you get this figure?

**Answer:** Figure-6B is the vertical deformation within glacial bodies obtained from the unwrapped phase information of the entire influence area. The lines in the manuscript has been modified as follows:

"However, because of the shape and size of the glacial bodies, it is difficult to unwrap glacial subset regions. Fig. 6B represents the displacement of glacial bodies (obtained from the unwrapped phase information for the entire sub-swath) where the glaciers lying close to the epicenter show positive displacement or uplift."

---

## Author Comment (AC3)

**Supplementary Information**

[Figure]

**Figure S1: Map showing (A) wrapped phase and (B) unwrapped phase within the influence area for 2020 Tibet earthquake (Mw 5.7).**

[Figure]

**Figure S2: USGS Shake Map for 2020 Tibet earthquake (Mw 5.7).**

[Figure]

**Supplementary Information**

Scale based on Worden et al. (2012)    Version 1: Processed 2022-01-25T17:06:17Z
△ Seismic Instrument  ○ Reported Intensity  ★ Epicenter

**Figure S3: USGS Isoseismal contours and the MMI chart for 2020 Tibet earthquake (Mw 5.7).**

**Table S1: Different focal mechanism solutions for the earthquake events considered in the study**

| Earthquake (Event ID) | Latitude (°) | Longitude (°) | Depth (km) | Nodal Plane-I | | | Nodal Plane-II | | | $M_w$ | Source |
|---|---|---|---|---|---|---|---|---|---|---|---|
| | | | | Strike-1 | Dip-1 | Rake-1 | Strike-2 | Dip-2 | Rake-3 | | |
| | 28.601 | 87.321 | 7 | 162 | 48 | 97 | 353 | 42 | 82 | 5.6 | IPGP |
| | 28.648 | 87.331 | 13 | 185.08 | 40.47 | 66.51 | 335.34 | 53.47 | 108.78 | 5.7 | GFZ |
| Tibet EQ (617805561) | 28.4065 | 87.2159 | 13.5 | 179.75 | 41.86 | 77.23 | 342.83 | 49.4 | 101.2 | 5.7 | NEIC |
| | 28.51 | 87.42 | 12 | 184 | 47 | 77 | 345 | 45 | 103 | 5.7 | GCMT |
| Leh EQ (619321061) | 34.208 | 78.205 | 10 | 27.62 | 40.6 | 94.12 | 213.04 | 49.53 | 86.48 | 5.1 | GFZ |
| Thang EQ (611835462) | 35.415 | 77.5847 | 80.5 | 87.05 | 45.63 | 100.39 | 252.36 | 45.32 | 79.55 | 5.2 | NEIC |
| | 35.74 | 77.45 | 118.1 | 118 | 65 | 74 | 332 | 30 | 120 | 5.2 | GCMT |

**Note: IPGP**- Institut de Physique du Globe de Paris; **GFZ**- German Research Centre for Geosciences; **NEIC**-National Earthquake Information Center; **GCMP**- Global Centroid-Moment-Tensor Project

**Supplementary Information**

[Figure]

**Figure S4: Beach ball representation for the focal mechanism solutions of the 2020 Tibet Earthquake (M$_w$ 5.7).**

**Supplementary Information**

[Figure]

**Figure S5: Map showing (A) wrapped phase and (B) unwrapped phase within the influence area for 2020 Leh earthquake (Mw 5.3).**

[Figure]

**Figure S6: Map showing (A) glacial and ground subset regions, (B) wrapped phase within ground subset, (C) unwrapped phase within ground subset, (D) wrapped phase within glacial subset and (E) unwrapped phase within glacial subset for 2020 Leh earthquake (Mw 5.3).**

**Supplementary Information**

[Figure]

**Figure S7: USGS Shake Map for 2020 Leh earthquake (Mw 5.3).**

**Supplementary Information**

[Figure]

**Figure S8: Beach ball representation for the focal mechanism solutions of the 2020 Leh Earthquake (M$_w$ 5.3).**

[Figure]

**Figure S9: Map showing (A) wrapped phase and (B) unwrapped phase within the influence area for 2017 Thang earthquake (Mw 5.2).**

**Supplementary Information**

[Figure]

**Figure S10: Map showing (A) glacial subset regions, (B) wrapped phase within glacial subset-1, (C) unwrapped phase within glacial subset-1, (D) wrapped phase within glacial subset-2 and (E) unwrapped phase within glacial subset-2 for 2017 Thang earthquake (Mw 5.2).**

[Figure]

**Figure S11: Map showing (A) ground subset regions, (B) wrapped phase within ground subset-1, (C) unwrapped phase within ground subset-1, (D) wrapped phase within ground subset-2 and (E) unwrapped phase within ground subset-2 for 2017 Thang earthquake (Mw 5.2).**

**Supplementary Information**

[Figure]

**Figure S12: USGS Shake Map for 2017 Thang earthquake (Mw 5.2).**

**Supplementary Information**

[Figure]

**Figure S13: Beach ball representation for the focal mechanism solutions of the 2017 Thang Earthquake (M$_w$ 5.2)**

[Figure]

**Supplementary Information**

**Figure S14: Map showing (A) wrapped phase and (B) unwrapped phase within the influence area for 2021 Joshimath earthquake (Mw 4.5).**

[Figure]

**Figure S15: Map showing (A) ground and glacial subset regions, (B) wrapped phase within glacial subset, (C) unwrapped phase within glacial subset, (D) wrapped phase within ground subset and (E) unwrapped phase within ground subset for 2021 Joshimath earthquake (Mw 4.5).**

**Supplementary Information**

[Figure]

**Figure S16: USGS Shake Map for 2021 Joshimath earthquake (Mw 4.5).**

[Figure]

**Figure S17: Map showing (A) wrapped phase and (B) unwrapped phase within the influence area for 2017 Sikkim earthquake (Mw 4.2).**

**Supplementary Information**

[Figure]

**Figure S18: Map showing wrapped phase (no information captured in unwrapping) within the influence area for 2018 Sikkim earthquake (Mw 4.4).**

[Figure]

**Figure S19: Map showing (A) wrapped phase and (B) unwrapped phase within the influence area for 2020 Nepal earthquake (Mw 4.1).**

**Supplementary Information**

[Figure]

**Figure S20: Map showing (A) wrapped phase and (B) unwrapped phase within the influence area for 2021 Nepal earthquake (Mw 4.1).**